# On the Expressivity and Sample Complexity of Node-Individualized Graph Neural Networks

**Paolo Pellizzoni**
Max Planck Institute of Biochemistry
Martinsried, Germany
`pellizzoni@biochem.mpg.de`

**Till Hendrik Schulz**
Max Planck Institute of Biochemistry
Martinsried, Germany
`tschulz@biochem.mpg.de`

**Dexiong Chen** [†]
Max Planck Institute of Biochemistry
Martinsried, Germany
`dchen@biochem.mpg.de`

**Karsten Borgwardt** [†]
Max Planck Institute of Biochemistry
Martinsried, Germany
`borgwardt@biochem.mpg.de`

## Abstract

Graph neural networks (GNNs) employing message passing for graph classification are inherently limited by the expressive power of the Weisfeiler-Leman (WL) test for graph isomorphism. Node individualization schemes, which assign unique identifiers to nodes (e.g., by adding random noise to features), are a common approach for achieving universal expressiveness. However, the ability of GNNs endowed with individualization schemes to generalize beyond the training data is still an open question. To address this question, this paper presents a theoretical analysis of the sample complexity of such GNNs from a statistical learning perspective, employing Vapnik–Chervonenkis (VC) dimension and covering number bounds. We demonstrate that node individualization schemes that are permutation-equivariant result in lower sample complexity, and design novel individualization schemes that exploit these results. As an application of this analysis, we also develop a novel architecture that can perform substructure identification (i.e., subgraph isomorphism) while having a lower VC dimension compared to competing methods. Finally, our theoretical findings are validated experimentally on both synthetic and real-world datasets.

## 1 Introduction

Graph Neural Networks (GNNs) have become a dominant approach in graph learning, leveraging the inherent structural information of graphs through multiple iterations of message passing. This iterative exchange of information between nodes allows GNNs to learn rich representations for both nodes and entire graphs. However, the expressivity of these models in the context of graph classification, i.e., their ability to distinguish non-isomorphic graphs, is intrinsically limited by the capabilities of the Weisfeiler-Leman (1-WL) algorithm [63], a heuristic used for the graph isomorphism problem [41, 64]. 1-WL is known to fail to distinguish certain classes of graphs, such as regular graphs [4]. Notably, GNNs are in general unable to solve the substructure identification task [12], which is concerned with identifying subgraph patterns in the data at hand.

To address these expressivity limitations, higher-order GNNs [41, 26] have been proposed. These models consider interactions among larger subgraphs or sets of nodes but often suffer from high

---

[†]These authors jointly supervised this work.

38th Conference on Neural Information Processing Systems (NeurIPS 2024).

computational complexity. An alternative approach, which we call *node individualization schemes*, entails introducing perturbations, such as random noise, into node features to break graph symmetries artificially. This technique enables message passing GNNs to become universal function approximators [1] without the computational overhead associated with higher-order GNNs. Several node individualization schemes have been proposed in the literature. Random Node Initializations (RNI) [52] add random noise to node labels to make them distinct with high probability, while Relational Pooling (RP) [42] and Colored Local Iterative Procedure (CLIP) [15] partition nodes based on their labels and assign unique identifiers within partitions. A recent paper [20] proposes a general framework based on individualization-refinement algorithms. Other approaches to increase the expressivity of GNNs relabel the graphs with structural or positional node encodings, such as node centralities, graphlet counts [9], Laplacian [16, 62, 28], and random walk [17] encodings. These encodings give information about the graph topology, complementing the information obtained from message passing, but cannot guarantee universal expressivity in general.

Beyond expressivity, the sample complexity of GNNs [36, 54, 23], i.e., the amount of training data required for generalization on unseen data, is a critical consideration to obtain effective graph learning models, and it has been recently linked to the 1-WL expressivity [37]. Nonetheless, the sample complexity of GNNs with node individualization schemes and with structural or positional node encodings, which is of paramount importance in the understanding of the generalization power of maximally expressive GNNs, remains an unanswered question.

This paper presents a novel theoretical analysis of both the expressivity and sample complexity of GNNs endowed with node individualization and structural or positional node encodings, bridging the gap between the expressivity analysis of GNNs and statistical learning theory. Specifically, we examine the sample complexity of various node individualization schemes in the binary graph classification setting, offering insights that can guide the design of more effective graph learning models. We make the following contributions:

- We demonstrate the utility of node individualization schemes even on graphs recognizable by the 1-WL test, thereby enhancing the expressivity of shallow GNNs (Section 3).
- We analyze the sample complexity of GNNs with various node individualization schemes using VC-dimension-based bounds (Theorem 1). Based on our analysis, we develop a model architecture EGONN that is universally expressive for substructure identification and has low VC dimension (Section 4.2).
- We provide a sharper analysis based on covering-number bounds, leading to the design of novel node individualization schemes (Theorem 4). Our bounds rely on a novel covering bound for Lipschitz functions [24], a result that is potentially of independent interest.
- We substantiate our theoretical findings with experimental results on both synthetic and real-world datasets (Section 5).

By offering a comprehensive analysis and practical guidance, this paper aims to enhance the understanding and development of GNNs through the lens of node individualization, pushing the boundaries of what these models can achieve in terms of expressivity and sample efficiency.

## 1.1 Related work

**Expressivity of GNNs** Since the seminal works of [41, 64] revealed the limitations of MPNNs due to their expressivity being bounded by the 1-WL test, research has surged toward developing more powerful GNNs. One prominent approach has been to design GNNs that emulate higher-order WL [26] or Folklore-WL [10] tests, exemplified by k-GNNs [41] and k-FGNNs [35]. However, their computational and memory demands often hinder real-world applicability. To mitigate this, researchers have explored more efficient alternatives, leveraging graph locality and sparsity [40, 67, 22]. Another line of research involves subgraph GNNs, which break symmetry by transforming the original graph into modified subgraphs for GNN processing [14, 48, 8]. For a more comprehensive overview, readers are encouraged to consult [51, 39].

**Node individualizations** Individualization schemes have been proposed by several works [52, 42, 15, 20, 11] to enhance the expressivity of GNNs. GNNs with such schemes have been shown to be universal function approximators for invariant functions [1, 5]. Individualization schemes are also crucial on edge-level tasks [34, 53], where they break symmetry among different edge-orbits.

**Sample complexity** The generalization capabilities of restricted families of GNNs has been addressed by means of VC dimension [54], Rademacher complexity [23] and PAC-bayesian approaches [29, 33]. Recently, [37, 21] proposed VC-dimension bounds based on the number of 1-WL color classes. The sample complexity of frequent subgraph identification has been addressed in [45]. Moreover, [46] and[19] address the sample complexity of equivariant models using a covering-number-based approach [6, 25], which is exploited also in [68] in an empirical setting. For a general overview on statistical learning theory, consult [36].

## 2 Preliminaries

In what follows, we define a graph as a tuple $G = (V_G, E_G, L_G)$, with $V_G = \{1, \ldots, |V_G|\}$ a finite set of nodes, and $E_G \subseteq \{\{u, v\} : u \neq v \in V_G\}$ a set of undirected edges. We define the vertex-label function as $L_G : V_G \to \mathcal{L}$, with a finite set of labels $\mathcal{L}$. For the sake of simplicity, we consider edges to be unlabeled. We define the neighborhood of a node as $\mathcal{N}(v) = \{w \in V_G : \{v, w\} \in E_G\}$. We say that two graphs $G$ and $H$ are isomorphic, denoted as $G \simeq H$, if there exists a bijective mapping $\pi : V_G \to V_H$, called isomorphism, such that $L_G(v) = L_H(\pi(v))$, $\forall v \in V_G$ and $\{\pi(u), \pi(v)\} \in E_H$ if and only if $\{u, v\} \in E_G$. The isomorphism relation induces equivalence classes, which we call *unordered* graphs [27]. The group of isomorphisms from $G$ to itself is called the automorphism group $Aut(G)$. We denote a set of graphs with $\mathcal{G}$, and if necessary, as $\mathcal{G}_\mathcal{L}$ to explicate the label set. We denote sets of unordered graphs as quotient sets $\mathcal{G}/\simeq$. Moreover, we consider sets of graphs of bounded size, that is $|V_G| \leq n_{\max}, \forall G \in \mathcal{G}$. A function defined on graphs $f : \mathcal{G} \to \mathbb{R}$ is called invariant if for any $G \simeq H$, $f(G) = f(H)$. A function from graphs to graphs $f : \mathcal{G} \to \mathcal{G}'$ is here called equivariant if for any $G \simeq H$, $f(G) \simeq f(H)$.

### 2.1 The Weisfeiler–Leman and Tinhofer algorithms

The color refinement algorithm, also known as 1-Weisfeiler–Leman (denoted as WL) algorithm, is a heuristic algorithm for the graph isomorphism problem. Let $C_0(v) = L_G(v) \in \mathbb{N}$ be the initial color of node $v \in V_G$. Then the algorithm updates vertex colors at iteration $k > 0$ as

$$C_k(v) = \text{HASH}\left(C_{k-1}(v), \{\{C_{k-1}(w) : w \in \mathcal{N}(v)\}\}\right) \in \mathbb{N},$$

with HASH an injective map. Let $\mathcal{P}^{(k)}(G)$ be the partitioning of $V_G$ based on colors after the $k$-th iteration of color refinement. Then, after at most $|V_G|$ iterations, the color partition stabilizes, i.e. $\mathcal{P}^{(k)}(G) = \mathcal{P}^{(k+1)}(G), \forall k \geq |V_G|$. Two graphs are deemed $k$-hop WL-isomorphic if $\{\{C_k(v) : v \in V_G\}\} = \{\{C_k(v) : v \in V_H\}\}$, and WL-isomorphic if it holds for $k = |V_G|$. We denote this as $G \simeq_{\text{WL}} H$. Note that $G \simeq H \implies G \simeq_{\text{WL}} H$, but the converse is not true. We call a graph $G$ *WL-amenable* if $\forall H$ such that $G \not\simeq H$, $G \not\simeq_{\text{WL}} H$.

Tinhofer [59] developed a color-refinement-based algorithm, described in Section A.2, that returns an ordering of the nodes of a graph. This is a canonical ordering on a large class of graphs, called Tinhofer, which is a strict superset of WL-amenable class [4]. The Tinhofer algorithm is an instance of the individualization-refinement paradigm [20] on which several isomorphism solvers are based [3].

### 2.2 Graph neural networks

Message passing graph neural networks (GNNs), given a graph $G$, iteratively produce for each node $v \in V_G$, at each level $k = 1, \ldots, K$, the embeddings $h_v^k \in \mathbb{R}^{d_k}$ by taking into account *messages* coming from its neighbors $\mathcal{N}(v)$. More formally, the embedding of node $v$ is updated as $h_v^k = f_{\text{upd}}\left(h_v^{k-1}, f_{\text{agg}}\left(\{\{h_u^{k-1} : u \in \mathcal{N}(v)\}\}\right)\right)$, where $f_{\text{agg}}$ and $f_{\text{upd}}$ are the aggregate and the update operations, respectively. The first layer of the GNN is fed with the initial node embeddings $h_v^0$, e.g. one-hot encodings of the node labels. Finally, one can get a graph-level readout $h_G \in [0, 1]$ by aggregating the output node embeddings via a function $f_{\text{out}}$. In [64, 41] it was shown that there exist injective functions $f_{\text{agg}}$, $f_{\text{upd}}$ and $f_{\text{out}}$ yielding GNNs that are provably as expressive as color refinement. We provide examples for such architectures in Section A.1.

We denote as $\text{GNN}_K = \{\text{GNN}_{K,\theta}, \theta \in \Theta\}$ the class of parametric functions formed by such a model with $K$ message passing layers and parameter space $\Theta$. Moreover, we define $\text{GNN}_{K,\theta}^{\text{bin}}(G) = \mathbb{1}\left[\text{GNN}_{K,\theta}(G) > 0.5\right]$ and $\text{GNN}_K^{\text{bin}} = \{\text{GNN}_{K,\theta}^{\text{bin}}, \theta \in \Theta\}$ the class of predictors.

## 3    Node individualization schemes

In this section, we present a general framework for node individualization schemes.

**Definition 1.** *A graph* $G = (V, E, L_G)$ *is called* individualized *if all node labels are pairwise distinct, that is* $|\{L_G(v) : v \in V\}| = |V|$. *Moreover, we define a graph as* $k$-weakly individualized *if all nodes have different colors after* $k$ *color refinement iterations, that is* $|\{C_k(v) : v \in V\}| = |V|$.

Graph individualization is effectively achieved by assigning unique identifiers to nodes through a relabeling process. We formally define a relabeling function $\mathrm{Rel} : \mathcal{G}_\mathcal{L} \times \Omega \to \mathcal{G}_{\mathcal{L}'}$ as a transformation $((V, E, L), \omega) \mapsto (V, E, L')$ that maps injectively the node labels from $L : V \to \mathcal{L}$ to a new set of labels $L' : V \to \mathcal{L}'$, e.g. with $\mathcal{L}' = \mathcal{L} \times \mathcal{C}$ and $L'(v) = (L(v), c)$, $c \in \mathcal{C}$. This function preserves the graph's underlying structure (the nodes and edges). For the sake of generality, the relabeling function incorporates an additional argument $\omega \in \Omega \subseteq \mathbb{N}$. This integer can be interpreted as a source of pseudo-randomness involved in the relabeling process (i.e., a random seed). For example, the relabeling function used in RNI [52], due to its pseudo-random nature, can generate non-isomorphic relabeled graphs even from two identical copies of the same graph if it is given two different random seeds. Moreover, we note that node encodings such as the random walk [17] or the Laplacian [16] positional encodings can be cast as relabeling functions.

A relabeling function is an *individualization scheme* if, for each $(G, \omega) \in \mathcal{G} \times \Omega$, it holds that $\mathrm{Rel}(G, \omega)$ is a (possibly $k$-weakly) individualized graph. Moreover, for $G \in \mathcal{G}$, let $\mathrm{Rel}(G) := \{\mathrm{Rel}(G, \omega), \omega \in \Omega\}$ and let $\mathrm{Rel}(\mathcal{G}) := \{\mathrm{Rel}(G) : G \in \mathcal{G}\}$.

Expressive GNNs on individualized graphs are well known to be universal approximators for functions over graphs [1]. In fact, even a GNN with a single message passing layer and a MLP head is enough to provide a universal approximator [15]. We can generalize this result to $k$-weakly individualized graphs by noticing that $k$ message passing layers, each simulating a WL iteration, provide a graph individualized by its node embeddings. The following statement formalizes this intuition.

**Proposition 1.** *Let* $f : \mathcal{G} \to \{0, 1\}$ *be an invariant function and* $k \geq 1$. *Then there exists* $\theta \in \Theta$ *such that* $\mathrm{GNN}_{k,\theta}^{\mathrm{bin}}(G) = f(G)$ *for every* $(k-1)$-*weakly individualized graph* $G \in \mathcal{G}$.

The above proposition highlights the importance of node individualization schemes, not only for enhancing model expressivity, but also for managing model complexity. While GNNs are theoretically capable of representing arbitrary functions on WL-amenable graphs (which compose the majority of graphs [4]), they might require a large number of message-passing layers to achieve this. This phenomenon, known as under-reaching [2], can be problematic. GNNs with many layers can suffer from oversmoothing [43] or oversquashing [2], hindering their ability to effectively distinguish graphs in practice. However, as Proposition 1 demonstrates, endowing GNNs with a (possibly $k$-weak) individualization scheme can offer sufficient signals for graph discrimination with fewer message-passing iterations, thus mitigating these issues. In Section 5.1, we experimentally investigate this assessment and show that it has implications in real-world scenarios.

## 4    Sample complexity bounds

In this section, we show a general framework to derive sample complexity bounds for GNNs endowed with relabeling functions, for graph-level binary classification tasks.

Let $\mathcal{D}_{\mathrm{data}}$ be a joint distribution over a set of graphs $\mathcal{G}$ and binary labels $Y = \{0, 1\}$. Moreover, let $\mathcal{D}$ be a joint distribution over the domain of $\mathcal{D}_{\mathrm{data}}$ and integers $\Omega \subseteq \mathbb{N}$. The integers $\omega \in \Omega$ model any (pseudo-)randomness within the classification model $\tilde{f}(G, \omega)$. In the following, we denote by $x = (G, \omega)$. Given a loss function $\ell(f(x), y)$, our goal, in accordance with standard learning theory [36], is to bound the difference between the true risk $R(f)$ and the empirical risk $\hat{R}(f)$:

$$R_\ell(f) = \mathbb{E}_{(x,y)\sim\mathcal{D}}[\ell(f(x), y)] \quad \text{and} \quad \hat{R}_{D,\ell}(f) = \frac{1}{m}\sum_{i=1}^{m}\ell(f(x_i), y_i),$$

where $D = \{(x_i, y_i)\}_i \sim \mathcal{D}^m$ represents a training dataset of size $m$ sampled i.i.d. from $\mathcal{D}$.

In Sections 4.1 and 4.2 we derive bounds to $|R_\ell(f) - \hat{R}_{D,\ell}(f)|$ based on the VC dimension, while in Section 4.3 we derive more refined bounds based on covering numbers.

## 4.1 VC dimension

Let $\ell(f(x), y) = \mathbb{1}[f(x) \neq y]$ be the 0-1 loss. Then if $d$ is the VC dimension of the hypothesis class, we have, with probability $1 - \delta$ over the dataset $D \sim \mathcal{D}^m$, that $\sup_{f \in \mathcal{F}} |R_\ell(f) - \hat{R}_{D,\ell}(f)| \leq \sqrt{2d/m} + O(\sqrt{\log(1/\delta)/m})$ [36, Corollary 3.4]. We define formally the VC dimension of a class of functions on a set.

**Definition 2.** *Let $X$ be a set and $\mathcal{F} \subseteq \{f : X \to \{0,1\}\}$ an hypothesis class. We say that $\mathcal{F}$ shatters $S = (x_1, \ldots x_{|S|})$ if $|\{(f(x_1), \ldots, f(x_{|S|})) : f \in \mathcal{F}\}| = 2^{|S|}$. Then, we say that $(X, \mathcal{F})$ has VC dimension $\mathrm{VC}(X, \mathcal{F}) = d$ if the largest set $S$ shattered by $\mathcal{F}$ has size $d$, and $+\infty$ if the size of sets that can be shattered is unbounded.*

In particular, we consider a set $X = \mathcal{G} \times \Omega$ consisting of pairs of graphs and integers. Our hypothesis class $\mathcal{F}$ comprises binary classification functions formed by the composition of graph relabeling functions (such as individualization schemes) and GNNs that are as expressive as color refinement: $\mathcal{F} = \mathrm{GNN}_K^{\mathrm{bin}} \circ \mathrm{Rel} := \{\mathrm{GNN}_{K,\theta}^{\mathrm{bin}} \circ \mathrm{Rel} : \mathrm{GNN}_{K,\theta}^{\mathrm{bin}} \in \mathrm{GNN}_K^{\mathrm{bin}}\}$, setting $K = n_{\max}$ to ensure expressivity. GNNs with color refinement expressivity may require a number of parameters exponential in the maximum graph size $n_{\max} = \max_{G \in \mathcal{G}} |V_G|$ [1], which is a strong assumption we have to make. In the cases where the parameter count constraints the expressive power, the VC bounds from [37, Theorem 6] can be applied instead. The following theorem, which follows from [37, Proposition 2], relates the VC dimension to the WL-isomorphism classes of the graphs in $\mathrm{Rel}(\mathcal{G})$.

**Theorem 1.** *Let $\mathrm{Rel} : \mathcal{G} \times \Omega \to \mathcal{G}'$ be a relabeling function. Then $\mathrm{VC}\left(\mathcal{G} \times \Omega, \mathrm{GNN}_K^{\mathrm{bin}} \circ \mathrm{Rel}\right) = |\mathrm{Rel}(\mathcal{G})/\simeq_{\mathrm{WL}}|$. If $\mathrm{Rel}$ is an individualization scheme, $|\mathrm{Rel}(\mathcal{G})/\simeq_{\mathrm{WL}}| = |\mathrm{Rel}(\mathcal{G})/\simeq|$.*

We provide specific bounds, proven in Section C.2, for relevant examples of relabeling schemes.

**Random Node Initializations [52]**  The RNI scheme perturbs with some random noise the labels of the nodes. In line with the analysis in the original paper, we suppose that the random noise, selected via $\omega$, takes values in a finite set $\mathcal{C}$. We then have that a graph $G$ with $n$ nodes is mapped to $O(|\mathcal{C}|^n/|Aut(G)|)$ relabeled unordered graphs. In the case of unlabeled graphs with $n$ nodes, we then have that $|\mathrm{RNI}(\mathcal{G})/\simeq| = \Theta(\frac{1}{n!} 2^{\binom{n}{2}} |\mathcal{C}|^n) = \Theta(|\mathcal{C}|^n \cdot |\mathcal{G}|)$.

**Relational Pooling [42] and CLIP [15]**  Let $G$ be a graph and let $v_1, \ldots, v_n$ be an arbitrary enumeration of its nodes selected via $\omega$. The simplest version of the RP scheme updates node labels as $\ell'(v_i) = (\ell(v_i), i)$. A more refined version of the scheme partitions the nodes as $\{V_1, \ldots, V_C\}$ based on their labels and, letting $V_c = (v_{i_1}, \ldots, v_{i_{|V_c|}})$, it updates node labels as $\ell'(v_{i_j}) = (\ell(v_{i_j}), j)$. Then, each graph can be mapped to $\prod_c |V_c|!/|Aut(G)|$ relabeled unordered graphs. For unlabeled graphs with $n$ nodes , this amounts to having $|\mathrm{RP}(\mathcal{G})/\simeq| = \Theta(2^{\binom{n}{2}}) = \Theta(n! \cdot |\mathcal{G}|)$. The 1-CLIP node individualization scheme is equivalent to RP on the node partition based on initial labels.

**Tinhofer algorithm**  Let $G$ be a graph and let $v_1, \ldots, v_n$ be an ordering given by the Tinhofer [59, 4] algorithm (Section A.2). Then updating node labels as $\ell'(v_i) = (\ell(v_i), i)$ yields a valid individualization scheme $\mathrm{Tinhofer}$, that is an instance of the general individualization-refinement framework given by [20]. Let the algorithm perform at most (i.e, for any choice of $\omega$) $I$ individualization iterations on graph $G$, let $\{V_1, \ldots, V_C\}$ be a partitioning of the nodes of $G$ based on their labels and let $R = \max_c |V_c|$. Then we have that $|\mathrm{Tinhofer}(G)/\simeq| \leq R^I/|Aut(G)| \leq n^I/|Aut(G)|$. In particular, as shown in the following lemma, for data distributions restricted to WL-amenable graphs, endowing a GNN with the $\mathrm{Tinhofer}$ individualization scheme does not increase its VC dimension.

**Lemma 1.** *Let $\mathcal{G}$ be a set of WL-amenable graphs. Then $\mathrm{VC}(\mathcal{G} \times \Omega, \mathrm{GNN}_K^{\mathrm{bin}} \circ \mathrm{Tinhofer}) = |\mathcal{G}/\simeq|$.*

In fact, the class of graphs for which the Tinhofer individualization scheme does not increase the VC dimension is the class of Tinhofer graphs [4] and is strictly larger than the WL-amenable class.

**Positional and structural encodings**  Deterministic and equivariant encodings such as the random walk structural encoding [17] or adding graphlet counts [9] yield $|\mathrm{Rel}(\mathcal{G})| = |\mathcal{G}|$, and therefore the increase in VC dimension is only due in the increase in the number of WL-isomorphism classes, if any. The Laplacian positional encoding [16] is known not to be equivariant in general [62],

i.e., there are graphs $G_1 \simeq G_2$ for which $\mathrm{Rel}(G_1, \omega) \not\simeq \mathrm{Rel}(G_2, \omega)$. Therefore we have that $|\mathrm{Rel}(\mathcal{G})/ \simeq_{WL} | > |\mathcal{G}/ \simeq_{WL} |$, which leads to an increase in the VC dimension and further motivates the attempts to obtain equivariant Laplacian-based relabeling functions [62, 28].

**Discussion** Note that, in general, the number of relabeled graphs given by $\mathrm{Rel}$ is lower if the relabeling scheme is equivariant and if it does not depend on the second argument $\omega$, i.e., if it is deterministic. For example, expressive GNNs endowed with the Tinhofer relabeling scheme, when applied to WL-amenable graphs with at most $n$ nodes, have VC dimension that is lower by a factor $O(n!)$ compared to equally expressive GNNs endowed with the RP scheme.

Finally, relabeling schemes that rely on the randomness introduced by the second argument $\omega$ (e.g. randomized schemes) are often enhanced by resampling different $\omega$ values at each epoch [42, 51], effectively performing data augmentation. As discussed in [56], when the augmented dataset includes all possible transformations of each data point (in this case, all possible graph relabelings), the VC dimension of the model reduces to that of a transformation-invariant model. However, the number of relabeled graphs in our setting often grows super-exponentially with the maximum graph size, e.g., $n_{\max}!$ for RP, making it difficult to obtain enough samples for good generalization, as shown experimentally in Section 5.2. The theoretical analysis of sample complexity for randomized individualization schemes remains an open question we leave for future work.

## 4.2 Improved VC bounds for substructure identification

In this section, we showcase that developing sample complexity bounds and relabeling schemes with low VC dimension can guide the design of architectures tailored to specific tasks. In particular, a relevant task that has been studied in past works is substructure identification [12, 28, 45], where the task is to decide if a graph pattern $P$ appears as a subgraph or induced subgraph of a query graph $G$, denoted hereafter as $P \in G$. Notably, message passing GNNs are unable to solve the task in general [12]. We provide a message-passing GNN architecture, similar to the one proposed in [12, Section 5] that, paired with node individualization schemes, is provably expressive for substructure identification and with lower VC dimension compared to the ones presented in Section 2.2. The idea, as in many subgraph GNNs [14, 48, 8, 50], is to transform the original graph into a set of smaller subgraphs and feeding them to a GNN.

In particular, let the task at hand concern a subgraph pattern $P$ of radius $\Delta_P = \min_{v \in V_P} \max_{w \in V_P} \delta_{vw}$, where $\delta_{vw}$ is the length of the shortest path from $v$ to $w$. Let a $k$-ego-net of node $v$ in graph $G$ be the induced subgraph $\mathrm{EGO}_{v,G,k}$ of nodes $\{u \in V_G : \delta_{uv} \leq k\}$ at distance at most $k$ from $v$. Then, a model that is able to recognize if $P$ appears in any of the $\Delta_P$-ego-nets of the query graph will be able to solve the task. Note that there could be ego-nets that are not recognizable by a GNN alone (Lemma 6), so we individualize each ego-net with a function $\mathrm{Rel}$ to guarantee expressivity. Let then

$$h_v^{\mathrm{ego}} = \mathrm{GNN}_{1,\theta}^{\mathrm{bin}} \left( \mathrm{Rel}(\mathrm{EGO}_{v,G,\Delta_P}, \omega) \right) \in \{0,1\}, \qquad h_G = \max_{v \in V_G} h_v^{\mathrm{ego}} \in \{0,1\},$$

that is, we run a 1-layer GNN on each individualized (potentially using some randomness from $\omega$) ego-net and we aggregate the results using max-pooling. We denote such parametric functions as $\mathrm{EGONN}_{\Delta_P,\theta}^{\mathrm{Rel}}$. We have the following results on the expressivity and the VC dimension of the model.

**Theorem 2.** *Let $f : \mathcal{G} \to \{0,1\}$ be $f(G) = \mathbb{1}[P \in G]$. Let $\mathrm{Rel}$ be an individualization scheme. Then there exists $\theta \in \Theta$ such that $\mathrm{EGONN}_{\Delta_P,\theta}^{\mathrm{Rel}}(G) = f(G)$ for every $G \in \mathcal{G}$.*

**Theorem 3.** *Let $\mathcal{G}_\Delta$ be the set of ego-nets of radius $\Delta$ of the graphs of $\mathcal{G}$. Then $\mathrm{VC}(\mathcal{G} \times \Omega, \mathrm{EGONN}_\Delta^{\mathrm{Rel}}) \leq |\mathrm{Rel}(\mathcal{G}_\Delta)/ \simeq |$.*

In general, $\mathcal{G}_\Delta$ is much smaller compared to $\mathcal{G}$, especially for small $\Delta$. This leads to the fact that, in general, $\mathrm{Rel}(\mathcal{G}_\Delta)$ will be much smaller than $\mathrm{Rel}(\mathcal{G})$. Thanks to Theorem 1, we then have that the VC dimension of $\mathrm{EGONN}_\Delta^{\mathrm{Rel}}$ is in general lower compared to the one of $GNN_K^{\mathrm{bin}} \circ \mathrm{Rel}$. In particular, relabeling the ego-nets using a scheme with lower $|\mathrm{Rel}(\mathcal{G}_\Delta)/ \simeq |$, such as the Tinhofer scheme, leads to lower VC dimension compared to schemes that would produce more relabeled ego-nets, such as RP or RNI. This theoretical result is also experimentally validated in Section 5.3.

This approach clearly increases the space complexity compared to the standard GNN model. Indeed, if a graph $G$ has $n$ nodes, and each ego net has at most $n_E$ nodes and $m_E$ edges, the disjoint union of the ego-nets of $G$ has $nn_E$ nodes and $nm_E$ edges.

## 4.3 Covering numbers

The generalization bounds based on the VC dimension theory provide results that are useful only in restricted cases, e.g., when the task is substructure identification. Indeed, for a maximally expressive GNN model the VC-based bounds state that one should observe all the possible individualized graphs from the data distribution before the model can generalize. We provide an alternative analysis, following a Rademacher-complexity-based approach introduced in [6] and used in [23] for graph classification, which can lead to tighter results.

Let the margin function be $M(f(x), y) = (2y - 1)(2f(x) - 1) \in [-1, 1]$. Then the margin loss, given a $\gamma > 0$, is defined as

$$\ell(f(x), y) = \mathbb{1}[M(f(x), y) < 0] + (1 - M(f(x), y)/\gamma)\mathbb{1}[0 \le M(f(x), y) \le \gamma] \in [0, 1].$$

Let $\mathcal{F}_\ell = \{(x, y) \to \ell(f(x), y) : f \in \mathcal{F}\}$ be the set of margin losses, for each predictor $f \in \mathcal{F} := \mathrm{GNN}_K \circ \mathrm{Rel}$. We then have [36], with probability $1 - \delta$ over the dataset $D = \{(x_i, y_i)\}_i \sim \mathcal{D}^m$, that $\sup_{f \in \mathcal{F}} |R_\ell(f) - \hat{R}_{D,\ell}(f)| \le \hat{\mathfrak{R}}_D(\mathcal{F}_\ell) + O(\sqrt{\ln(1/\delta)/m})$, with $\hat{\mathfrak{R}}_D(\mathcal{F}_\ell)$ the empirical Rademacher complexity of $\mathcal{F}_\ell$ on $D$ (see Definition 3 in the Appendix). In particular, given that $\mathbb{P}_{(x,y) \sim \mathcal{D}}[\mathbb{1}[f(x) > 0.5] \ne y] \le R(f)$, one can then bound the probability of making an error at inference time from the empirical loss. We then bound the empirical Rademacher complexity $\hat{\mathfrak{R}}_D(\mathcal{F}_\ell)$ via a covering-number-based approach. Let $D$ be the training dataset and let $\mathcal{F}_\ell|_D = \{f|_D : f \in \mathcal{F}_\ell\}$ be the set of margin losses restricted to the dataset.

Given a pseudometric space $(X, \mathrm{dist})$ and a subset $S \subseteq X$, we call $C \subseteq X$ an $\epsilon$-cover of $S$ if $\max_{x \in S} \min_{c \in C} \mathrm{dist}(x, c) \le \epsilon$ and define the $\epsilon$-covering number of $S$ as $\mathcal{N}(X, \epsilon, \mathrm{dist}) = \min\{|C| : C \text{ is an } \epsilon\text{-cover of } S\}$, i.e., the minimum number of balls of radius $\epsilon$ required to cover $S$. Similarly, for a finite $S$, we call $C \subseteq X$ a $p$-norm $\epsilon$-cover of $S$ if $\left(1/|X| \sum_{x \in X} \min_{c \in C} \mathrm{dist}(x, c)^p\right)^{1/p} \le \epsilon$ and define the $p$-norm $\epsilon$-covering number of $S$ as $\mathcal{N}^{(p)}(X, \epsilon, \mathrm{dist}) = \min\{|C| : C \text{ is a } p\text{-norm } \epsilon\text{-cover of } S\}$. With slight abuse of notation, we let $\mathcal{N}^{(\infty)}(X, \epsilon, \mathrm{dist}) = \mathcal{N}(X, \epsilon, \mathrm{dist})$. In particular, we have that the $p$-norm $\epsilon$-covering number of a metric space is monotonically increasing in $p \in \mathbb{N} \cup \{\infty\}$.

Moreover, for a finite set $X$ and $\mathcal{F} \subseteq \{f : X \to [0, 1]\}$, we define the pseudometrics $\|f_1 - f_2\|_p = \left(1/|X| \sum_{x \in X} |f_1(x) - f_2(x)|^p\right)^{1/p}$ for $p \ge 1$ and $\|f_1 - f_2\|_\infty = \max_{x \in X} |f_1(x) - f_2(x)|$. By Pollard's bound [47] we then have that

$$\hat{\mathfrak{R}}_D(\mathcal{F}_\ell) \le \inf_{\alpha > 0} \left(\alpha + \sqrt{2 \log \mathcal{N}(\mathcal{F}_\ell|_D, \alpha, \|\cdot\|_1)/|D|}\right).$$

We bound the covering number of the class of functions $\mathcal{F}_\ell$ by covering the space of graphs and assuming the Lipschitzness of GNNs, similarly to the approaches in [19] and [46]. In fact, we provide a generalization of a covering bound for Lipschitz functions [24, Lemma 5.2], that relaxes the Kolmogorov-Tikhomirov bound [58, Eq. 238], tailored to the aforementioned Rademacher bound.

**Lemma 2.** *Let $(X, \mathrm{dist})$ be a pseudometric space and $S \subseteq X$ a finite subset. Let $\mathcal{F} \subseteq \{f : S \to [0, 1]\}$ be a set of $C$-Lipschitz functions. Let $p, q \in \mathbb{N} \cup \{+\infty\}, q \ge p \ge 1$. Then*

$$\log \mathcal{N}(\mathcal{F}, \alpha, \|\cdot\|_p) \le \log(1/\alpha + 1) \cdot \mathcal{N}^{(q)}\left(S, \frac{\alpha}{2C}, \mathrm{dist}\right).$$

**Theorem 4.** *Let $\mathcal{F}_\ell$ be the set of margin losses for predictors $\mathcal{F} := \mathrm{GNN}_K \circ \mathrm{Rel}$. Let $\mathrm{Rel}(D) = \{G' = \mathrm{Rel}(G, \omega) : (G, \omega, y) \in D\}$ be the graphs of the dataset after the relabeling, endowed with a pseudometric $\mathrm{dist}$. Let functions in $\mathrm{GNN}_K$ be $C$-Lipschitz continuous with respect to $\mathrm{dist}$. Then*

$$\hat{\mathfrak{R}}_D(\mathcal{F}_\ell) \le \inf_{\alpha > 0} \left(\alpha + |D|^{-1/2} \sqrt{2 \log(1/\alpha + 1) \cdot 4 \mathcal{N}^{(1)}\left(\mathrm{Rel}(D), \frac{\alpha \gamma}{4C}, \mathrm{dist}\right)}\right).$$

Theorem 4 relies on the assumption that the GNN models are Lipschitz with respect to some pseudometric $\mathrm{dist}$ on graphs, such as the edit distance [55, 44], the Tree Mover distance [13] or the WWL distance [60]. This assumption, albeit strong, is often taken when considering the generalization of GNNs [28]. Note that, for small $\alpha$ and arbitrarily complex GNNs, i.e. $C \gg 1$, we have that the covering number is proportional to the number of WL-isomorphic graphs in the relabeled dataset $\mathrm{Rel}(D)$, effectively retrieving the bounds obtained by the VC dimension. When $\mathrm{GNN}_K$ is instead simpler, i.e. it has smaller Lipschitz constant $C$, the space of graphs can be covered with fewer balls, and we obtain better sample complexity.

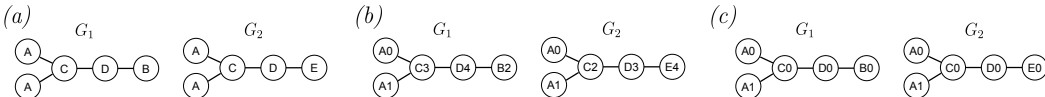

Figure 1: Comparison between Tinhofer and Tinhofer$_W$. *Panel (a):* Two graphs, where letters indicate initial node labels. The Tinhofer algorithm finds a canonical ordering on the two graphs. *Panel (b):* The Tinhofer scheme concatenates the position of the node in the ordering to the node label. The relabeled graphs have edit distance 3. *Panel (c):* The Tinhofer$_W$ scheme concatenates the position of the node within its WL color class. The edit distance remains 1, as in the original graphs.

**Discussion**   Let the graphs of the dataset $D_G = \{G : (G, \omega, y) \in D\}$ and the relabeled graphs $\mathrm{Rel}(D) = \{\mathrm{Rel}(G, \omega) : (G, \omega, y) \in D\}$ be subsets of the same metric space $(\mathcal{G}, \mathrm{dist})$. The results of Theorem 4 suggest that if a labeling function does not increase much the covering number of the dataset, i.e. $\mathcal{N}^{(1)}(\mathrm{Rel}(D), \epsilon, \mathrm{dist}) \simeq \mathcal{N}^{(1)}(D_G, \epsilon, \mathrm{dist})$, then the model $\mathrm{GNN}_K \circ \mathrm{Rel}$ would have roughly the same sample complexity as $\mathrm{GNN}_K$ while being more expressive.

In particular, this can be achieved if the relabeling scheme maps pairs of graphs with low distance (e.g. $\mathrm{dist}(G_1, G_2) \leq \epsilon$) to pairs of relabeled graphs with low distance. An example of such graphs at low distance could be a pair of graphs that differ only in the label of a node belonging to a singleton color class, like graphs $G_1$ and $G_2$ in Figure 1. These two graphs could be, for example, two molecules where an atom is substituted with a similar one, or two proteins where an amino-acid is substituted with a functionally-similar one [44].

Driven by this intuition, we design a ($k$-weak) individualization scheme, which we call *weak Tinhofer* scheme $\mathrm{Tinhofer}_W$, that relabels only few nodes. The scheme obtains an ordering $v_1, \ldots, v_n$ of the nodes via the Tinhofer algorithm. Let $\{V_1, \ldots, V_C\}$ be a partition of the nodes of graph $G$ into color classes after $k$ color refinement steps and let $V_c = (v_{i_1}, \ldots, v_{i_{|V_c|}})$. The scheme then relabels nodes by appending their position in the ordering within their partition as $\ell'(v_{i_j}) = (\ell(v_{i_j}), j)$. For nodes that belong to singleton classes, which is common in graphs such as molecules or random graphs, no relabeling is necessary.

See Figure 1 for a visual example of a comparison between Tinhofer and Tinhofer$_W$ on the pair of graphs $G_1$ and $G_2$. We have, for a $u \in V_G$, that $\exists \phi : V_{G_1} \to V_{G_2}$ such that $\{\phi(v), \phi(w)\} \in E_{G_2}$ iff $\{v, w\} \in E_{G_1}$, $L_{G_2}(\phi(v)) = L_{G_1}(v), \forall v \neq u \in V_{G_1}$ and $L_{G_2}(\pi(u)) \neq L_{G_1}(u)$. Then $\mathrm{dist}(G_1, G_2)$ would be given by the cost of substituting $L_{G_2}(\pi(u))$ with $L_{G_1}(u)$. Consider the graphs produced by the Tinhofer$_W$ procedure. Here we have that the labels of nodes that matched in the original graphs still match in the relabeled graphs. By contrast, with the graphs produced by the Tinhofer procedure, the number of node labels that do not match could be arbitrarily large.

This scheme ensures that the relabeled graphs are $k$-weakly individualized, while yielding no increase in the VC dimension for WL-amenable graphs (Lemma 1) and increasing the covering numbers only marginally. This in general yields better generalization, as shown experimentally in Section 5.4.

## 5   Experimental evaluation

In this section, we provide proof-of-concept experimental evidence about our theoretical results. In particular, Section 5.1 investigates the empirical expressivity of GNNs with and without individualization schemes, Section 5.2 and Section 5.3 validate respectively the general VC dimension bounds and the substructure identification ones, and Section 5.4 explores the results on covering numbers. Details about the datasets, experimental setup, tasks and infrastructure are reported in Section E. Code and datasets are available at `https://github.com/BorgwardtLab/NodeIndividualizedGNNs`.

### 5.1   Expressivity

To highlight the efficacy of individualization schemes in practical applications, we compare a GNN endowed with Tinhofer to ordinary GNNs on challenging datasets. In particular, we consider subsets of the real-world datasets `MCF-7` and `Peptites-func`, for which at least six message-passing iterations are necessary to distinguish all pairs of graphs, and synthetic datasets `Cycles-pin` and `CSL-pin`, for which nine, resp. five, iterations are necessary. To compare the performance between

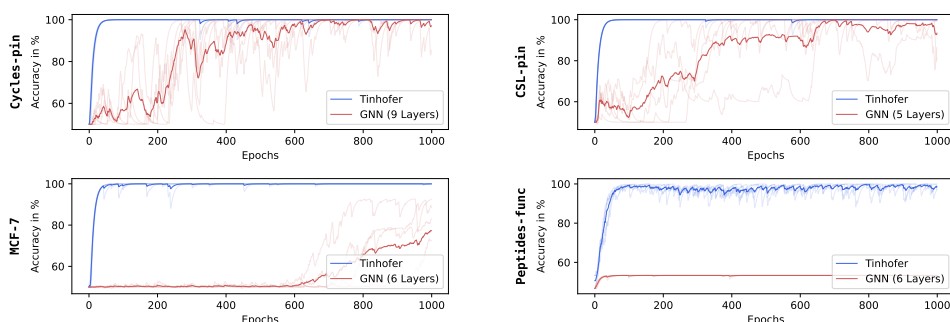

Figure 2: Accuracy for $\mathrm{GNN}_K$ and $\mathrm{GNN}_1 \circ \mathrm{Tinhofer}$ over 1000 epochs on synthetic datasets `Cycles-pin` and `CSL-pin` and real-world datasets `MCF-7` [65, 38] and `Peptites-func` [57, 18].

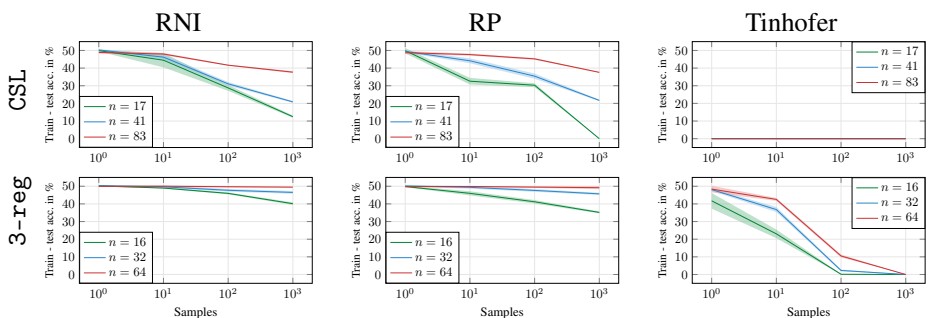

Figure 3: Difference between test and training accuracy for GNNs with the RNI, RP and Tinhofer individualization schemes, on datasets of CSL and 3-regular graphs of various sizes.

the two methods, we measure the models' capabilities to distinguish non-isomorphic graphs. Figure 2 shows the accuracy obtained by ordinary GNNs with a theoretically sufficient number of layers as well as a 1-layer GNN endowed with Tinhofer. While the ordinary GNNs can achieve a perfect score for both synthetic datasets, their learning process is slow and unstable. For the real-world datasets, it can be observed that ordinary GNNs do not achieve a score close to $100\%$ within the tested 1000 epochs at all. These observations may be explained by the fact that the model has to learn to propagate structural information for several layers, possibly encountering oversquashing issues. In contrast, the GNN with Tinhofer converges faster and more stably with only a single GNN layer. The results particularly emphasize that the Tinhofer individualization scheme can improve the expressive power in shallow GNNs, demonstrating the value of our approach in practical scenarios.

## 5.2 VC dimension

We empirically evaluate the generalization bounds based on the VC dimension derived in Section 4.1. Recall that for an individualization scheme Rel and a set of graphs $\mathcal{G}$, the generalization gap is bounded by the VC dimension $d = |\mathrm{Rel}(\mathcal{G})/ \simeq |$. We evaluate RNI, RP, and Tinhofer using datasets of circular skip link (CSL) [42] graphs of sizes $n = 17, 41, 83$ and 3-regular graphs of sizes $n = 16, 32, 64$. The task is to distinguish non-isomorphic graphs. Figure 3 shows the generalization gap with increasing numbers of permuted copies of the graphs in the training set. RNI and RP, which generate highly randomized individualizations, require several samples to generalize effectively. As graph size $n$ increases, the VC dimension increases, resulting in a greater generalization gap. Interestingly, Tinhofer perfectly learns the target function on CSL graphs with only a single copy of the graphs, as the individualization is in fact canonical.

## 5.3 Substructure identification

To assess the expressivity and generalization of GNNs for substructure identification, we created four challenging datasets, each comprising 1000 unlabeled 3-regular graphs. The task is to detect whether

graphs contain the induced subgraphs: 3-cycle $C_3$, 4-cycle $C_4$, 5-cycle $C_5$, or complete bipartite graph $K_{2,3}$. We tested GNNs with and without individualization schemes, along with our novel architecture $\text{EGONN}_{\Delta_P}$. As reported in Table 1, GNNs lacking individualization fail to distinguish between graphs. Those with RP or Tinhofer individualizations tend to overfit on training data but struggle to generalize. In contrast, $\text{EGONN}_{\Delta_P}$ models generally exhibit a smaller generalization gap. Not providing individualizations to ego-nets results in a small generalization gap, but often fails to fit the training data. RP individualization enables the model to fit training data but may lead to overfitting. The Tinhofer individualization yields, as predicted by the theory, the best performance.

Table 1: Train and test accuracies for the substructure identification task on 3-regular graphs.

| GNN | Rel | Datasets | | | | | | | |
| | | $C_3$ | | $C_4$ | | $C_5$ | | $K_{2,3}$ | |
| | | Train | Test | Train | Test | Train | Test | Train | Test |
|---|---|---|---|---|---|---|---|---|---|
| $\text{GNN}_K$ | None | $50.1_{\pm 0.0}$ | $49.0_{\pm 0.0}$ | $50.6_{\pm 0.0}$ | $45.0_{\pm 0.0}$ | $50.1_{\pm 0.0}$ | $49.0_{\pm 0.0}$ | $50.0_{\pm 0.0}$ | $50.0_{\pm 0.0}$ |
| | RP | $100.0_{\pm 0.0}$ | $59.8_{\pm 2.8}$ | $100.0_{\pm 0.0}$ | $60.6_{\pm 2.6}$ | $100.0_{\pm 0.0}$ | $50.4_{\pm 1.7}$ | $100.0_{\pm 0.0}$ | $59.6_{\pm 3.6}$ |
| | Tinhofer | $100.0_{\pm 0.0}$ | $62.0_{\pm 3.3}$ | $100.0_{\pm 0.0}$ | $61.4_{\pm 3.9}$ | $100.0_{\pm 0.0}$ | $47.0_{\pm 2.1}$ | $100.0_{\pm 0.0}$ | $62.0_{\pm 4.0}$ |
| $\text{EGONN}_{\Delta_P}$ | None | $100.0_{\pm 0.0}$ | $100.0_{\pm 0.0}$ | $74.2_{\pm 7.2}$ | $73.6_{\pm 10.2}$ | $92.3_{\pm 1.9}$ | $89.4_{\pm 1.5}$ | $74.1_{\pm 2.9}$ | $74.2_{\pm 2.2}$ |
| | RP | $100.0_{\pm 0.0}$ | $100.0_{\pm 0.0}$ | $99.6_{\pm 0.5}$ | $93.2_{\pm 2.1}$ | $99.4_{\pm 0.6}$ | $90.6_{\pm 1.0}$ | $97.0_{\pm 0.9}$ | $71.6_{\pm 1.9}$ |
| | Tinhofer | $100.0_{\pm 0.0}$ | $100.0_{\pm 0.0}$ | $99.4_{\pm 0.6}$ | $99.6_{\pm 0.5}$ | $99.8_{\pm 0.3}$ | $99.6_{\pm 0.8}$ | $100.0_{\pm 0.0}$ | $100.0_{\pm 0.0}$ |

## 5.4 Covering numbers

To validate the theoretical results on covering numbers from Section 4.3, we use two molecular (NCI1 and Mutagenicity) and two social network (IMDB-b and COLLAB-b) datasets. As suggested by the theory, we use as a proxy for the sample complexity the ratio between the covering number (for simplicity fixing $\epsilon = 0.05$, see Section F for complete results) of the relabeled dataset and the dataset size $\hat{\mathcal{N}} = \mathcal{N}^{(1)}(\text{Rel}(D), \epsilon, \text{dist})/|D|$. As a (pseudo-)metric, we use the distance used by the WWL kernel [60]. We report $\hat{\mathcal{N}}$, together with the test accuracy and the difference between test and train accuracy, in Table 2 for GNNs with no individualizations, with RP, RNI, Tinhofer and weak Tinhofer individualizations, and with a Laplacian-encoding-based relabeling. We observe that across datasets $\hat{\mathcal{N}}$ strongly correlates with the generalization gap, confirming empirically the intuition provided by the theory. Moreover, we observe that on molecular datasets, that are easily recognizable by the WL test, individualizations don't help expressivity. On the other hand, on unlabeled datasets such as COLLAB-b, ordinary GNNs might struggle to converge to good local minima, and the additional information given by relabeling schemes usually helps. In fact, it is common in the literature [64, 41] to relabel such unlableled graphs with, e.g., degree centrailities.

Table 2: Covering numbers and classification accuracies on real-world datasets.

| GNN | Rel | Datasets | | | | | | | | | | | |
| | | NCI1 | | | Mutagenicity | | | IMDB-b | | | COLLAB-b | | |
| | | $\mathcal{N}$ | Test | Diff. | $\mathcal{N}$ | Test | Diff. | $\mathcal{N}$ | Test | Diff. | $\mathcal{N}$ | Test | Diff. |
|---|---|---|---|---|---|---|---|---|---|---|---|---|---|
| $\text{GNN}_5$ | None | 0.090 | $83.8_{\pm 0.5}$ | $16.1_{\pm 0.5}$ | 0.156 | $83.0_{\pm 2.6}$ | $17.0_{\pm 2.6}$ | 0.025 | $71.8_{\pm 5.0}$ | $3.6_{\pm 5.9}$ | 0.214 | $56.1_{\pm 3.1}$ | $1.1_{\pm 2.9}$ |
| | RP | 0.719 | $67.8_{\pm 1.5}$ | $32.2_{\pm 1.5}$ | 0.746 | $70.9_{\pm 1.3}$ | $29.1_{\pm 1.3}$ | 0.443 | $65.4_{\pm 2.6}$ | $33.0_{\pm 2.5}$ | 0.488 | $67.5_{\pm 9.1}$ | $3.1_{\pm 2.5}$ |
| | RNI | 0.705 | $67.1_{\pm 1.8}$ | $32.9_{\pm 1.8}$ | 0.717 | $73.1_{\pm 1.5}$ | $26.9_{\pm 1.5}$ | 0.580 | $62.4_{\pm 9.0}$ | $37.6_{\pm 9.0}$ | 0.547 | $56.8_{\pm 3.6}$ | $1.2_{\pm 2.9}$ |
| | Tinhofer | 0.677 | $71.5_{\pm 2.2}$ | $28.4_{\pm 2.2}$ | 0.697 | $74.9_{\pm 0.7}$ | $25.1_{\pm 0.7}$ | 0.266 | $70.0_{\pm 2.6}$ | $19.1_{\pm 2.7}$ | 0.471 | $64.8_{\pm 7.0}$ | $2.0_{\pm 3.4}$ |
| | Tinhofer$_w$ | 0.203 | $81.9_{\pm 1.2}$ | $18.0_{\pm 1.2}$ | 0.282 | $82.6_{\pm 1.4}$ | $17.4_{\pm 1.4}$ | 0.133 | $67.8_{\pm 4.0}$ | $21.3_{\pm 4.0}$ | 0.310 | $79.5_{\pm 2.6}$ | $13.6_{\pm 3.2}$ |
| | LPE | 0.589 | $77.3_{\pm 1.8}$ | $22.6_{\pm 1.8}$ | 0.485 | $76.2_{\pm 1.4}$ | $23.8_{\pm 1.4}$ | 0.271 | $68.0_{\pm 3.3}$ | $21.1_{\pm 3.4}$ | 0.409 | $74.6_{\pm 2.4}$ | $21.6_{\pm 2.7}$ |
| $\text{GNN}_2$ | None | 0.090 | $81.8_{\pm 1.4}$ | $18.1_{\pm 1.5}$ | 0.156 | $81.5_{\pm 1.3}$ | $18.4_{\pm 1.3}$ | 0.025 | $71.4_{\pm 3.9}$ | $2.2_{\pm 4.4}$ | 0.214 | $75.3_{\pm 1.1}$ | $0.0_{\pm 1.0}$ |
| | RP | 0.719 | $67.4_{\pm 1.2}$ | $32.6_{\pm 1.2}$ | 0.746 | $71.2_{\pm 1.0}$ | $28.8_{\pm 1.0}$ | 0.443 | $63.8_{\pm 2.3}$ | $34.6_{\pm 2.3}$ | 0.488 | $75.1_{\pm 2.3}$ | $18.1_{\pm 3.1}$ |
| | RNI | 0.705 | $68.0_{\pm 1.0}$ | $32.0_{\pm 1.0}$ | 0.717 | $72.7_{\pm 2.4}$ | $27.3_{\pm 2.4}$ | 0.580 | $63.6_{\pm 8.8}$ | $36.4_{\pm 8.8}$ | 0.547 | $72.1_{\pm 2.8}$ | $19.6_{\pm 4.9}$ |
| | Tinhofer | 0.677 | $72.9_{\pm 3.0}$ | $26.9_{\pm 3.0}$ | 0.697 | $73.1_{\pm 2.3}$ | $26.9_{\pm 2.3}$ | 0.266 | $68.6_{\pm 3.1}$ | $20.5_{\pm 3.1}$ | 0.471 | $75.2_{\pm 1.7}$ | $19.5_{\pm 1.2}$ |
| | Tinhofer$_w$ | 0.203 | $81.8_{\pm 2.3}$ | $18.1_{\pm 2.4}$ | 0.282 | $81.2_{\pm 1.7}$ | $18.8_{\pm 1.7}$ | 0.133 | $69.4_{\pm 3.4}$ | $19.6_{\pm 3.2}$ | 0.310 | $80.8_{\pm 1.9}$ | $12.2_{\pm 2.1}$ |
| | LPE | 0.589 | $76.4_{\pm 1.9}$ | $23.5_{\pm 1.9}$ | 0.485 | $75.4_{\pm 2.0}$ | $24.6_{\pm 2.0}$ | 0.271 | $68.4_{\pm 3.3}$ | $20.7_{\pm 3.3}$ | 0.409 | $75.8_{\pm 1.8}$ | $19.9_{\pm 2.3}$ |

## 6 Conclusion

In this paper, we developed novel sample complexity bounds for graph neural networks endowed with individualization schemes. Several research directions are left open for future work, including the analysis of data augmentation for individualization schemes via resampling, and the development of tighter covering bounds for specific model architectures. We envision that this work will inspire the development of new, practical graph learning models that are both theoretically sound and empirically effective, ultimately advancing the field of graph representation learning.

## Acknowledgements

The authors would like to thank Dr. Janko Sattler, Philip Hartout and the reviewers for their feedback.

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

# A  Additional details

## A.1  Expressive graph neural networks

In [64] it was shown that there exist injective functions for $f_{\mathrm{agg}}$, $f_{\mathrm{upd}}$ and the graph-level readout function that lead to GNNs that are provably as expressive as color refinement.

An example of such functions that leads to models that are provably as expressive as color refinement [41], denoting $\|$ as concatenation, is

$$h_v^k = \mathrm{mlp}\Big(h_v^{k-1}\Big\| \sum_{u\in\mathcal{N}(v)} h_u^{k-1}\Big) \in \mathbb{R}^{d_k} \qquad h_G = \mathrm{mlp}\Big(\sum_{v\in V_G} h_v^K\Big) \in [0,1].$$

In fact, provided that different node labels are encoded to linearly independent $h_v^0$'s, even the following simpler architecture, denoting with $\sigma$ a nonlinear function such as ReLU, is as expressive as color refinement [41]:

$$h_v^k = \sigma\Big(W_1^k h_v^{k-1} + W_2^k \sum_{u\in\mathcal{N}(v)} h_u^{k-1}\Big) \in \mathbb{R}^{d_k} \qquad h_G = \mathrm{mlp}\Big(\sum_{v\in V_G} h_v^K\Big) \in [0,1].$$

## A.2  Tinhofer algorithm

The Tinhofer algorithm [59, 4] returns an ordering of the nodes of a graph. In particular, it works as follows.

1. Run color refinement on $G$ and obtain the stable color partition $\mathcal{P}(G)$.

2. Given the partition $\mathcal{P}(G)$

   - If all nodes belong to a singleton color class, return the ordering of the nodes based on the lexicographic order of their colors.
   - Else, pick the color class with at least two nodes with the lexicographically smallest color. Individualize one arbitrary node in such class by assigning it the smallest unused color. Then, go to step 1.

Clearly, the algorithm will perform at most $|V_G|$ individualization steps. In fact, for several classes of graphs a constant number $I$ of individualizations suffices [20]. Therefore, the algorithm can run in polynomial time.

In fact, the Tinhofer algorithm can be generalized to the Individualization-Refinement scheme [20, 3], where the refinement operation and the selection of the nodes to be individualized can be more general than the ones used in Tinhofer.

Computing a canonical stable partition on a graph $G$ can be done in $O((|V_G| + |E_G|)\log|V_G|)$ [7, Theorem 9]. Moreover, the Tinhofer algorithm can also be implemented so that it takes $O((|V_G| + |E_G|)\log|V_G|)$ time [7, Theorem 10].

# B  Limitations

This paper provides sample complexity results for graph neural networks with relabeling schemes, but nonetheless leaves several research directions open. First, most individualization schemes benefit from resampling at random the graph individualizations at each epoch, effectively performing data augmentation. Studying the theoretical properties of such techniques would be a major step forward in understanding the practical performance of GNNs with individualization schemes. Secondly, our results on covering numbers are based on the strong assumption of Lipschitzness of GNN models with respect to some metric on graphs. Alleviating such assumptions would yield stronger theoretical results. We hope that these open questions will foster new research efforts in this field.

# C Definitions and Lemmas

In this section, we provide formal definitions and lemmas, which were only informally described in the main paper.

## C.1 Definitions

**Definition 3.** *Let $X$ be a set and $S = (x_1, \ldots, x_n)$ a sample of $n$ elements from $X$. Let $\mathcal{F} \subseteq \{f : X \longrightarrow [0,1]\}$ be a family of functions. Let also $\sigma_i, i = 1, \ldots, n$ be independent and identically distributed Rademacher random variables, i.e. taking value $1$ or $-1$ with probability $1/2$. The* empirical Rademacher complexity *is the quantity*

$$\hat{\mathfrak{R}}_S(\mathcal{F}) = \mathbb{E}_\sigma \left[ \sup_{f \in \mathcal{F}} \frac{1}{n} \sum_{i=1}^n \sigma_i f(x_i) \right]$$

*where $\mathbb{E}_\sigma$ denotes the expectation taken only with respect to the $\sigma_i$'s, conditionally on the sample $S$.*

## C.2 Lemmas from Section 4.1

**Lemma 3.** *Let $\mathrm{RNI}$ be the RNI individualization scheme [52] with random noise taking values in $\mathcal{C}$ and $G$ a graph with $n$ nodes. Then $|\mathrm{RNI}(G)/ \simeq | = O(|\mathcal{C}|^n / |Aut(G)|)$. If $\mathcal{G}$ is the set of of unlabeled graphs with $n$ nodes, then $|\mathrm{RNI}(\mathcal{G})/ \simeq | = \Theta(\frac{|\mathcal{C}|^n}{n!} 2^{\binom{n}{2}})$.*

*Proof.* In line with the analysis in the original paper, we suppose that the random noise, selected via $\omega$, takes values in a finite set $\mathcal{C}$. Each node of a graph $G$ can take $|\mathcal{C}|$ values. Therefore there are $O(|\mathcal{C}|^n)$ different labelings of the graph. To obtain the number of unordered graphs, we divide by the number of automorphisms of $G$. Moreover, the set $\mathcal{G}$ of unlabeled graphs with $n$ nodes contains $\Theta(\frac{1}{n!} 2^{\binom{n}{2}})$ unordered graphs. Intuitively, is obtained from the fact that for each of the $\binom{n}{2}$ pairs of nodes one can either have an edge or not, and by then dividing by the number of node permutations. Formally, this is obtained via Polya's enumeration [27]. A fraction of $1 - o(1)$ of unordered graphs only have the trivial automorphism [27]. Therefore, we have the claim by summing a contribution of $O(|\mathcal{C}|^n)$ over all such graphs. $\square$

**Lemma 4.** *Let $\mathrm{RP}$ be the RP individualization scheme [42] and $G$ a graph with partitioning $\{V_1, \ldots, V_C\}$ of the nodes based on their labels. Then $|\mathrm{RP}(G)/ \simeq | = (\prod_{c=1}^C |V_c|!)/|Aut(G)|$. If $\mathcal{G}$ is the set of of unlabeled graphs with $n$ nodes, then $|\mathrm{RP}(\mathcal{G})/ \simeq | = \Theta(2^{\binom{n}{2}})$.*

*Proof.* Each partition $V_i$ can take $|V_i|!$ different orderings, for a total of $\prod_{c=1}^C |V_c|!$ different labelings of the graph. To obtain the number of unordered graphs, we divide by the number of automorphisms of $G$. Moreover, the set $\mathcal{G}$ contains $\Theta(\frac{1}{n!} 2^{\binom{n}{2}})$ unordered graphs, and a fraction of $1 - o(1)$ of those only have the trivial automorphism [27]. Therefore, we have the claim by summing a contribution of $\Theta(n!)$, i.e., the number of possible different labelings of each graph, over all such graphs. $\square$

**Lemma 5.** *Let $\mathrm{Tinhofer}$ be the Tinhofer individualization scheme and $G$ a graph with partitioning $\{V_1, \ldots, V_C\}$ of the nodes based on their labels. Let the algorithm perform at most (i.e, for any choice of $\omega$) $I$ individualization iterations on graph $G$, let $\{V_1, \ldots, V_C\}$ be a partitioning of the nodes of $G$ based on their labels and let $R = \max_c |V_c|$. Then we have that $|\mathrm{Tinhofer}(G)/ \simeq | \leq R^I/|Aut(G)| \leq n^I/|Aut(G)|$.*

*Proof.* After each individualization and color refinement iteration, the color class partition of the nodes of $G$ is a refinement of the partitioning $\{V_1, \ldots, V_C\}$ of the nodes of $G$ based on their labels. At each individualization step, the algorithm can choose to individualize a node in the lexicographically-smallest stable color class, which has at most $R = \max_c |V_c|$ nodes. Therefore we have at most $R^I$ labelings of the graph and we divide by the number of automorphisms to obtain the number of distinct unordered graphs. In particular $|\mathrm{Tinhofer}(G)| \leq R^I/|Aut(G)| \leq n^I/|Aut(G)|$. $\square$

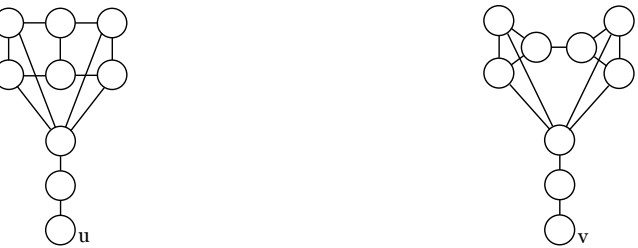

Figure 4: The gadgets $P_1$ and $P_2$ from Lemma 6

### C.3  Lemmas from Section 4.2

**Lemma 6.** *Let* Id *be the identity relabeling function. There exists a pattern $P$ and pairs of graphs $G_1 \not\simeq G_2$ such that $P \in G_1$, $P \notin G_2$ and such that they cannot be assigned different outputs by* $\mathrm{EGONN}^{Id}_{\Delta_P}$.

*Proof.* The graphs $P_1$ and $P_2$ depicted in Figure 4 are non-isomorphic. Let the pattern of interest be $P = P_1$. Then $\Delta_P = 2$. Consider a graph $G$. Let $G_1$ be the graph obtained by connecting all nodes of $G$ to the node $u$ of $P_1$ and $G_2$ by connecting all the nodes of $G$ to node $v$ of $P_2$, making sure to select $G$ such that $P \notin G_2$. Meanwhile, we have that $P \in G_1$ by construction.. Then for any $\Delta_P$-ego-net $E_1$ of $G_1$ there is a $\Delta_P$-ego-net $E_2$ of $G_2$ such that $E_1 \simeq_{\mathrm{WL}} E_2$. Therefore, $\mathrm{EGONN}^{Id}_{\Delta_P}$ will assign the same outputs to both $G_1$ and $G_2$. $\qquad\square$

### C.4  Lemmas from Section 4.3

**Lemma 7** (Pollard's bound [47])**.** *Let $X$ be a finite set and $\mathcal{F} \subseteq \{f : X \to [0,1]\}$. Consider the pseudometric on $\mathcal{F}$ defined as $\|f_1 - f_2\|_1 = 1/|X| \sum_{x \in X} |f_1(x) - f_2(x)|$. Then*

$$\hat{\mathfrak{R}}_X(\mathcal{F}) \le \inf_{\alpha > 0} \left( \alpha + \sqrt{2 \log \mathcal{N}(\mathcal{F}, \alpha, \| \cdot \|_1)/|X|} \right).$$

*Proof.* Let $C \subset \{f : X \to [0,1]\}$ be an $\alpha$-cover of $(\mathcal{F}, \| \cdot \|_1)$ of size $\mathcal{N}(\mathcal{F}, \alpha, \| \cdot \|_1)$. Let also $c_f \in C$ be such that $\|f - c_f\|_1 \le \alpha$. Then

$$\hat{\mathfrak{R}}_X(\mathcal{F}) = \mathbb{E}\left[ \sup_{f \in \mathcal{F}} \frac{1}{|X|} \sum_{x_i \in X} \sigma_i f(x_i) \right] \le$$

$$\le \mathbb{E}\left[ \sup_{f \in \mathcal{F}} \frac{1}{|X|} \sum_{x_i \in X} \sigma_i(f(x_i) - c_f(x_i)) \right] + \mathbb{E}\left[ \sup_{f \in \mathcal{F}} \frac{1}{|X|} \sum_{x_i \in X} \sigma_i c_f(x_i) \right] \le$$

$$\le \alpha + \mathbb{E}\left[ \sup_{c \in \mathcal{C}} \frac{1}{|X|} \sum_{x_i \in X} \sigma_i c(x_i) \right] \le \alpha + \sup_{c \in \mathcal{C}} \sqrt{\frac{2 \log |C|}{|X|}},$$

where the last inequality follows from Massart lemma and the fact that the functions in $C$ take values in $[0,1]$. $\qquad\square$

## D  Missing proofs

In this section, we provide the proofs that were omitted from the main paper.

### D.1  Proofs from Section 3

**Proposition 1.** *Let $f : \mathcal{G} \to \{0,1\}$ be an invariant function and $k \ge 1$. Then there exists $\theta \in \Theta$ such that $\mathrm{GNN}^{\mathrm{bin}}_{k,\theta}(G) = f(G)$ for every $(k-1)$-weakly individualized graph $G \in \mathcal{G}$.*

*Proof.* We consider the class of GNNs described in Section A.1, with depth $K = \max_{G \in \mathcal{G}} |V_G|$ and large enough width [41]. Let the initial node embedding $h_v^0$ of $v \in V_G$ be a one-hot encoding for its label $L_G(v)$, for any $G \in \mathcal{G}$. Applying [41, Theorem 2] to the disjoint union of the graphs in $\mathcal{G}$, we have that there exists a function $\mathrm{GNN}_{k-1,\theta} \in \mathrm{GNN}_{k-1}$ can simulate $k-1$ steps of color refinement for all graphs in $\mathcal{G}$. Therefore, from the definition of $(k-1)$-weakly individualized graph, after $(k-1)$ message passing layers, all the node embeddings $h_v^{k-1} \in \{-1,1\}^d$ of nodes $v \in V_G$ computed by $\mathrm{GNN}_{k-1,\theta}$ will be distinct, that is, the graph is individualized. Then this individualized graph is WL amenable and 1 color refinement iteration suffices to identify the graph [4]. Then by [37, Proposition 9], any invariant binary function on $\mathcal{G}$ can be realized. $\qquad\square$

## D.2  Proofs from Section 4.1

**Theorem 1.** *Let* $\mathrm{Rel} : \mathcal{G} \times \Omega \to \mathcal{G}'$ *be a relabeling function. Then* $\mathrm{VC}\left(\mathcal{G} \times \Omega, \mathrm{GNN}_K^{\mathrm{bin}} \circ \mathrm{Rel}\right) = |\mathrm{Rel}(\mathcal{G})/ \simeq_{\mathrm{WL}} |$. *If* $\mathrm{Rel}$ *is an individualization scheme,* $|\mathrm{Rel}(\mathcal{G})/ \simeq_{\mathrm{WL}} | = |\mathrm{Rel}(\mathcal{G})/ \simeq |$.

*Proof.* We consider the class of GNNs described in Section A.1, with depth $K = \max_{G \in \mathcal{G}} |V_G|$ and large enough width [41]. Let the initial node embedding $h_v^0$ of $v \in V_G$ be a one-hot encoding for its label $L_G(v)$, for any $G \in \mathcal{G}$.

Let $d := |\mathrm{Rel}(\mathcal{G})/ \simeq_{\mathrm{WL}} |$. We prove the upper bound, following the argument of [37, Proposition 8]. Let by contradiction $S$ be a (multi)set of $d+1$ elements that can be shattered. Then by pigeon hole, two elements $(G,\omega), (G',\omega') \in S$ must be mapped by $\mathrm{Rel}$ to graphs belonging to the same WL-isomorphism class. Then, by equivalence of $\mathrm{GNN}_K^{\mathrm{bin}}$ to color refinement [41, Theorem 1], these two graphs are always mapped to the same value by $\mathrm{GNN}_K^{\mathrm{bin}} \circ \mathrm{Rel}$. Then, there is no set of $d+1$ elements that can be shattered.

We then prove the lower bound. Let $S'$ be a set of $d$ graphs that are representatives of the WL-isomorphism classes contained in $\mathrm{Rel}(\mathcal{G})/ \simeq_{\mathrm{WL}}$. For $G' \in S'$, let $\mathrm{Rel}^{-1}(G') = (G,\omega) \in \mathcal{G} \times \Omega$ such that $\mathrm{Rel}(G,\omega) = G'$. Note that, since $\mathrm{Rel}$ is injective in the first argument, the graph $G$ is unique. On the other hand, $\omega$ may not be unique. We take any such $\omega$ with a deterministic rule (e.g. the minumum). Moreover, since we restrict the domain to $S'$, this inverse always exists. Let then $S = \{\mathrm{Rel}^{-1}(G') : G' \in S'\}$. We have that $|S| = |S'| = d$ and it can be shattered by [37, Proposition 9].

When the graphs in $\mathrm{Rel}(\mathcal{G})$ are individualized, WL-isomorphism class correspond to isomorphism classes [4], which proves the second claim. $\qquad\square$

**Lemma 1.** *Let* $\mathcal{G}$ *be a set of WL-amenable graphs. Then* $\mathrm{VC}\left(\mathcal{G} \times \Omega, \mathrm{GNN}_K^{\mathrm{bin}} \circ \mathrm{Tinhofer}\right) = |\mathcal{G}/ \simeq |$.

*Proof.* Tinhofer algorithm yields a canonical ordering on WL amenable graphs [4], therefore we have $|\mathrm{Tinhofer}(G)| = 1$, $\forall G \in \mathcal{G}$. Then, by Theorem 1 we have the claim. $\qquad\square$

## D.3  Proofs from Section 4.2

**Theorem 2.** *Let* $f : \mathcal{G} \to \{0,1\}$ *be* $f(G) = \mathbb{1}\left[P \in G\right]$. *Let* $\mathrm{Rel}$ *be an individualization scheme. Then there exists* $\theta \in \Theta$ *such that* $\mathrm{EGONN}_{\Delta_P, \theta}^{\mathrm{Rel}}(G) = f(G)$ *for every* $G \in \mathcal{G}$.

*Proof.* By Proposition 1 we have that $GNN_1 \circ \mathrm{Rel}$ is universally expressive. In particular, it can take value 1 if $P \in \mathrm{EGO}_{v,G,\Delta_P}$ and 0 otherwise for each possible $\mathrm{EGO}_{v,G,\Delta_P}$. Then $h_G = \max_{v \in V_G} h_v^{\mathrm{ego}} = 1$ if and only if $P \in G$. $\qquad\square$

**Theorem 3.** *Let* $\mathcal{G}_\Delta$ *be the set of ego-nets of radius* $\Delta$ *of the graphs of* $\mathcal{G}$. *Then* $\mathrm{VC}(\mathcal{G} \times \Omega, \mathrm{EGONN}_\Delta^{\mathrm{Rel}}) \leq |\mathrm{Rel}(\mathcal{G}_\Delta)/ \simeq |$.

*Proof.* We prove the upper bound. Let $d = |\mathrm{Rel}(\mathcal{G}_\Delta)/ \simeq |$. Suppose by contradiction that there is a set $G_1, \ldots G_{d+1}$ of $d+1$ graphs that can be shattered. In particular, each graph $G_i$ must have an ego net $H_i \in G_i$ that does not appear in any other graph, i.e. $H_i \notin G_j, \forall j \neq i$. Indeed, suppose that for

each ego-net $H_{v,i} = \text{EGO}_{v,G_i,\Delta}$ of graph $G_i$, we have that $H_{v,i} \in G_j$ for some $j \neq i$. Then to be able to assign $h_{G_i} = 1$, it must hold that $\text{GNN}_{1,\theta}(H_{v,i}) = 1$ for some $v$. But since $H_{v,i} \in G_j$, we have that $h_{G_j} = 1$ and the assignment of $G_j$ to 1 and all the other graphs to 0 cannot be achieved. Therefore, each graph $G_i$ must have an ego-net $H_i \in G_i$ that does not appear in any other graph. Then there must be $d+1$ distinct ego-nets, which is a contradiction. $\qquad\square$

**Remark 1.** *In fact, there are sets of graphs $\mathcal{G}$ and radii $\Delta$ for which the VC bound above is almost tight. This lower bound can be proven by taking a datasets of $k$ star graphs with maximum degrees $2, \ldots k+1$ and $\Delta = 1$. Each graph has an ego-net that is unique to the graph, i.e., the graph itself, so the set can be shattered. Moreover, $|\text{Rel}(\mathcal{G}_\Delta)/\simeq| = k+1$.*

### D.4 Proofs from Section 4.3

**Lemma 2.** *Let $(X, \text{dist})$ be a pseudometric space and $S \subseteq X$ a finite subset. Let $\mathcal{F} \subseteq \{f : S \to [0,1]\}$ be a set of $C$-Lipschitz functions. Let $p, q \in \mathbb{N} \cup \{+\infty\}, q \geq p \geq 1$. Then*

$$\log \mathcal{N}(\mathcal{F}, \alpha, \|\cdot\|_p) \leq \log(1/\alpha + 1) \cdot \mathcal{N}^{(q)}\left(S, \frac{\alpha}{2C}, \text{dist}\right).$$

*Proof.* We construct a covering of the space $\mathcal{F}$. Let $p \neq \infty$. Let $S_c = \{x_1, \ldots x_r\}$ a $p$-norm $\alpha/2C$-covering of $(S, \text{dist})$ of size $r = \mathcal{N}^{(p)}\left(S, \frac{\alpha}{2C}, \text{dist}\right)$. For a point $x \in S$, let its center be $c(x) = \text{argmin}_{x_i \in S_c} \text{dist}(x, x_i)$. Recall that a $p$-norm $\epsilon$-covering $C$ of a finite set $S$ is such that $\left(1/|S| \sum_{x \in S} \text{dist}(x, c(x))^p\right)^{1/p} \leq \epsilon$.

Let $\bar{\mathcal{F}} = \{\bar{f} : S_c \to \{(k + 1/2)\alpha : k = 0, \ldots, \lfloor \alpha^{-1} \rfloor\}\}$. For $\bar{f}_i \in \bar{\mathcal{F}}$, we construct an associated function $\hat{f}_i : S \to \{(k + 1/2)\alpha : k = 0, \ldots, \lfloor \alpha^{-1} \rfloor\}$ such that $\hat{f}_i(x) = \bar{f}_i(c(x)), \forall x \in S$. Then let $\hat{\mathcal{F}}$ be the set of maps $\{\hat{f} : \bar{f} \in \bar{\mathcal{F}}\}$. We have that $|\hat{\mathcal{F}}| \leq (1/\alpha + 1)^r$, as the functions can take up to $(1/\alpha + 1)$ values over $r$ points, and $|\bar{\mathcal{F}}| = |\hat{\mathcal{F}}|$. We show that $\hat{\mathcal{F}}$ is indeed an $\alpha$-covering of $\mathcal{F}$ in norm $\|\cdot\|_p$.

Let $f \in \mathcal{F}$. Let $\hat{f} \in \hat{\mathcal{F}}$ be such that $\forall x \in S_c, |\hat{f}(x) - f(x)| \leq \alpha/2$, which always exists by construction. Then,

$$\|f - \hat{f}\|_p = \left(1/s \sum_{x \in S} |f(x) - \hat{f}(x)|^p\right)^{1/p} \leq$$

$$\leq \left(1/s \sum_{x \in S} \left(|f(c(x)) - \hat{f}(c(x))| + |f(x) - f(c(x))| + |\hat{f}(x) - \hat{f}(c(x))|\right)^p\right)^{1/p}$$

$$\leq \left(1/s \sum_{x \in S} (\alpha/2 + C\text{dist}(x, c(x)) + 0)^p\right)^{1/p} \leq \left(\alpha^p/2 + C^p 2^{p-1} \cdot 1/s \sum_{x \in S} \text{dist}(x, c(x))^p\right)^{1/p}$$

$$\leq \left(\alpha^p/2 + C^p 2^{p-1} \cdot (\alpha/2C)^p\right)^{1/p} = \alpha.$$

In particular, we used that fact that $(a + b)^p \leq 2^{p-1}(a^p + b^p)$ for $p \geq 1$, which follows from Jensen's inequality with $f(x) = x^p$.

For $p = \infty$, we have that $S_c$ is a $\alpha/2C$-covering of $(S, \text{dist})$ of size $r = \mathcal{N}\left(S, \frac{\alpha}{2C}, \text{dist}\right)$, and $\max_{x \in S} \text{dist}(x, c(x)) \leq \alpha/2C$. We build $\hat{\mathcal{F}}$ as before. We then have

$$\|f - \hat{f}\|_\infty = \max_{x \in S} |f(x) - \hat{f}(x)| \leq \max_{x \in S} |f(c(x)) - \hat{f}(c(x))| + |f(x) - f(c(x))| + |\hat{f}(x) - \hat{f}(c(x))| \leq$$

$$\leq \alpha/2 + C\text{dist}(x, c(x)) + 0 \leq \alpha.$$

Finally, noting that $\mathcal{N}^{(q)}\left(S, \frac{\alpha}{2C}, \text{dist}\right) \geq \mathcal{N}^{(p)}\left(S, \frac{\alpha}{2C}, \text{dist}\right)$ for each $q \geq p$, we have the claim. $\qquad\square$

**Theorem 4.** *Let $\mathcal{F}_\ell$ be the set of margin losses for predictors $\mathcal{F} := \text{GNN}_K \circ \text{Rel}$. Let $\text{Rel}(D) = \{G' = \text{Rel}(G, \omega) : (G, \omega, y) \in D\}$ be the graphs of the dataset after the relabeling, endowed with a pseudometric $\text{dist}$. Let functions in $\text{GNN}_K$ be $C$-Lipschitz continuous with respect to $\text{dist}$. Then*

$$\hat{\mathfrak{R}}_D(\mathcal{F}_\ell) \leq \inf_{\alpha > 0}\left(\alpha + |D|^{-1/2}\sqrt{2\log(1/\alpha + 1) \cdot 4\mathcal{N}^{(1)}\left(\mathcal{G}_{n,\mathcal{L}'}, \frac{\alpha\gamma}{4C}, \text{dist}\right)}\right).$$

*Proof.* We have that $\hat{\mathfrak{R}}_D(\mathcal{F}_\ell) \leq \inf_{\alpha > 0} \left( \alpha + \sqrt{2 \log \mathcal{N}(\mathcal{F}_\ell|_D, \alpha, \|\cdot\|_1)/|D|} \right)$ by Lemma 7. We bound the covering number of the loss functions.

Let $\mathrm{Rel}(D) = \{\mathrm{Rel}(G, \omega) : (G, \omega, y) \in D\}$ be the graphs of the dataset after the relabeling, endowed with a pseudometric $\mathrm{dist}$. We endow $D$ with a pseudometric $\mathrm{dist}_D$ defined as $\mathrm{dist}((G, \omega, y), (G', \omega', y')) = \mathrm{dist}_D(\mathrm{Rel}(G, \omega), \mathrm{Rel}(G', \omega'))$ if $y = y'$ and $+\infty$ otherwise. Moreover, if functions in $\mathrm{GNN}_K$ are $C$-Lipschitz on $\mathrm{Rel}(D)$ then the functions in $\mathcal{F}$ are $C$-Lipschitz on $D$ by construction. Then, functions $\ell(f(x), y) \in \mathcal{F}_\ell$ are $2C/\gamma$-Lipschitz on $D$.

Let also $r$ be the 1-norm covering number of $\mathrm{Rel}(D)$ and $S$ the set of $r$ points that cover the set. Then we can obtain a cover of $(D, \mathrm{dist}_D)$ of size $2r$ by taking the points $\{(x, 1), (x, 0) : x \in S\}$.

We can then apply Lemma 2 and obtain that

$$\log \mathcal{N}(\mathcal{F}_\ell|_D, \alpha, \|\cdot\|_1) \leq \log(1/\alpha + 1) \cdot 2\mathcal{N}^{(q)}\left(\mathrm{Rel}(D), \frac{\alpha\gamma}{4C}, \mathrm{dist}\right).$$

Plugging this result in the first formula, we obtain the claim. $\qquad\square$

# E  Experimental Setups and Datasets

In this section, we provide detailed descriptions to the experimental setups and datasets used in Sect. 5.

## E.1  Computing infrastructure

The experiments are run on a cluster equipped with Intel(R) Xeon(R) Silver 4116 CPUs and NVIDIA H100 GPUs. The code is based on PyTorch and PyTorch-Geometric.

## E.2  Datasets

In the following, we describe and discuss the datasets and their properties. Real-world datasets (i.e., NCI1, IMDB-b, MCF-7, Mutagenicity, COLLAB-b, and Peptides-func) were provided by [31, 38] and [18]. A summary of all datasets and their properties can be found in Tab. 3.

**CSL** We include three circular skip link (CSL) datasets generated analogously to the setup considered in [42]. A CSL graph is a 4-regular graph $G_{n,s}$ with $n > 4$ and $1 < s < n/2$ which consists of a cycle of length $n$ on which all pairs of nodes at distance $s$ on this cycle are connected by an additional edge. For a given $n$, we generate a dataset by considering the graphs $\{G_{n,s} : s \in S\}$ where $S$ is the set of all skip values leading to non-isomorphic graphs. In particular, we consider graph sizes $n \in \{17, 41, 83\}$ for which there exist $4, 10$, and $20$ non-isomorphic graphs, respectively. We refer to the datasets as CSL-17, CSL-41, and CSL-83. Since classification tasks in this work are generally restricted to binary classification, we assign the graphs of a CSL dataset evenly into two classes. The classification task is then to assign graphs to their binary skip link class. This setup is commonly used as a benchmark for testing the expressive power of a model since 4-regular graphs cannot be distinguished by the 1-WL test.

**3-Reg** 3-Reg are datasets consisting of 3-regular graphs which are common types of expressivity benchmark datasets. For each $n \in \{16, 32, 64\}$, we generated a dataset containing $10$ randomly generated 3-regular graphs of size $n$. We refer to these three datasets as 3-reg-16, 3-reg-32, and 3-reg-64. We evenly distributed the 10 graphs into two groups such that the binary classification task is to identify the group that a graph belongs to.

**Cycles-pin** As an example dataset containing $k$-weakly individualized graphs, we generated 10 graphs of size 33. Each such graph $G_i$ consists of a pair of cycles of sizes $16 - i$ and $16 + i$ for $i \in \{0, \ldots, 9\}$. We then append a pin, i.e., an additional node, to one of the nodes in the smaller of the two cycles. We evenly distributed the 10 graphs into two groups. The binary classification task is then to identify the group that a graph belongs to.

Table 3: Overview of real-world and synthetic graph dataset properties. Datasets marked (*) are composed of subsets of the original datasets. The column "WL number" denotes the minimum number of WL iterations necessary to distinguish all graphs. The entry (-) in this column indicates that this number does not exist, i.e., not all graphs can be distinguished by color refinement.

| Name | # Graphs | Classes | Avg. # nodes | Avg. # edges | Node labels | WL number |
|---|---|---|---|---|---|---|
| COLLAB-b | 5000 | 2 | 74.49 | 2457.78 | − | 1 |
| IMDB-b | 1000 | 2 | 19.77 | 96.53 | − | 1 |
| MCF-7* | 14 | 2 | 45.86 | 50.29 | + | 6 |
| Mutagenicity | 4337 | 2 | 30.32 | 30.77 | + | 3 |
| NCI1 | 4110 | 2 | 29.87 | 32.30 | + | 3 |
| Peptites-func* | 15 | 2 | 169.47 | 172.93 | + | 6 |
| 3-reg-16 | 10 | 2 | 16 | 24 | − | − |
| 3-reg-32 | 10 | 2 | 32 | 48 | − | − |
| 3-reg-64 | 10 | 2 | 64 | 96 | − | − |
| CSL-17 | 4 | 2 | 17 | 34 | − | − |
| CSL-41 | 10 | 2 | 41 | 82 | − | − |
| CSL-83 | 20 | 2 | 83 | 166 | − | − |
| CSL-pin | 10 | 2 | 42 | 83 | − | 5 |
| Cycles-pin | 10 | 2 | 33 | 33 | − | 9 |

**CSL-pin**   Analogously to the Cycles-pin dataset, we generate a dataset called CSL-pin by appending a pin, i.e. an additional node, to one of the nodes in every graph of the CSL-41 datasets described above. Adding the pins turns the graphs into $k$-weakly individualized graphs.

**COLLAB-b [66]**   COLLAB consists of 5,000 graphs extracted from three scientific collaboration networks. We merge two of the three classes (class 2 and 3) to make the labels binary. Each graph corresponds to an ego-network which represents the collaboration network of a particular researcher. The graph is then annotated by the field of this researcher such that the classification task consists of assigning an ego-network to the research field.

**IMDB-b [66]**   IMDB-b includes 1,000 ego-network graphs obtained from movie collaboration networks. In such graphs, nodes represent actors and edges represent their co-appearance in a movie. An ego-network graph is then classified by the genre of the movie collaboration network it was extracted from.

**MCF-7**   From the original MCF-7 [65, 38] dataset, we extracted a maximal set of graphs (up to isomorphism) such that no graph can be distinguished from all others by the Weisfeiler-Lehman (WL) test after four iterations. We divided these 14 graphs into two groups such that the binary classification tasks, i.e. assigning a graph to its group, requires at least five color refinement iterations to improve over random guessing and six iterations to perfectly classify all graphs.

**Mutagenicity [30, 49]**   Mutagenicity is a binary classification benchmark dataset consisting of 4,337 molecular graphs. The classification task is to predict the mutagenic property of a molecule.

**NCI1 [61]**   NCI1 is a binary dataset containing 4,110 molecular compounds, each categorized based on their ability to inhibit the growth of a non-small cell lung cancer cell line.

**Peptides-func**   From the original Peptides-func [57, 18] dataset, we extracted a maximal set of graphs (up to isomorphism) such that no graph can be distinguished from all others by the Weisfeiler-Lehman (WL) test after four iterations. We divided these 15 graphs into two groups such that the binary classification tasks, i.e. assigning a graph to its group, requires at least five color refinement iterations to improve over random guessing and six iterations to perfectly classify all graphs.

### E.3 Experimental setup

In the following, we give detailed descriptions of the experimental setups of our evaluation section in the main paper.

#### E.3.1 Expressivity

For the expressivity evaluation, we considered the datasets `Cycles-pin`, `CSL-pin`, `MCF-7`, and `Peptides-func` which are described in detail in Sect. E.2. The experiments compare the capability of ordinary GNNs with that of a 1-layer GNN that is run on top of the Tinhofer individualization to learn to distinguish WL-distinguishable graphs. For each dataset, we provide the training accuracies obtained by a $\text{GNN}_K$, where $K$ is the minimum number of color refinement iterations necessary to distinguish all graphs in the respective dataset, as well as the training accuracies obtained by $\text{GNN}_1 \circ \text{Tinhofer}$. We use, as the $\text{GNN}_K$ architecture, the one described in Section A.1, with $K$ MLP-based message passing layers. In fact, the GNNs based on linear layers for message passing, despite their theoretical equivalence to color refinement, were unable to converge, further strengthening our claims.

The plotted results in Fig. 2 show the accuracy over 1000 epochs where we depict the average over a total of 5 runs in bold and the individual runs faded out. We fixed the embedding dimension to 256 and use an Adam optimizer with a learning rate of 0.0001. Due to instability in the learning process, we provide smoothened learning curves for better clarity of the results.

#### E.3.2 VC dimension

To investigate the generalization bounds based on the VC-dimension results, we compare the individualization methods RNI [52], RP [42], and Tinhofer. The experiments are conducted on the CSL and 3-Reg datasets of varying numbers of nodes. To investigate the generalization performance, we generate as the training set $1, 10, 100$, and $1000$ permutations for each graph in the respective dataset and test on 1000 permuted copies. We report the performance in terms of the difference between the train and test accuracy. Particularly, for every dataset, we consider the generalization gap achieved at the last epoch with a perfect training accuracy score over a total of 1000 epochs. We conduct a total of 5 runs and plot the mean as well as the standard deviations over these runs.

In all cases, we used the $\text{GNN}_K$ architecture described in Section A.1, with $K = 4$ linear-layer-based message passing layers. We fix the embedding dimension to 256 and use an Adam optimizer with a learning rate of 0.0001.

#### E.3.3 Substructure identification

For the substructure identification task , we generate 1000 3-regular graphs of order 30 for each of the task using NETWORKX. In particular, 500 graphs contain the pattern $P$ and 500 do not. We split the dataset in train and test by assigning 90% of the graphs to the train set at random.

We use, as the $\text{GNN}_K$ architecture, the one described in Section A.1, with $K$ linear-layer-based message passing layers. In particular, $K$ is set to 2 for $C_3, C_4, K_{2,3}$ and to 3 for $C_5$, as these are the minimum number of layers for the message passing to fully cover these patterns.

We use the same architecture GNN for the $\text{EGONN}_{\Delta_P}$ model on each of the $\Delta_P$-ego-nets, with 1 layer for the individualized versions and $\Delta_P$ layers for the non-individualized version. In particular, $\Delta_P$ is 1 for $C_3$ and 2 for $C_4, C_5, K_{2,3}$. Moreover, we replace the max aggregation with a softmax, in order to allow the gradients to be propagated more easily, as we observe that this allows for an easier convergence. We fix dimension of the GNN embeddings to 256. We use the Adam optimizer [32] with learning rate 0.001 and batch size 256.

The model is trained for 1000 epochs. The epoch at which the accuracies are reported is selected as the one with highest training accuracy, with ties broken by the lowest training loss. Note that since the focus of this paper is on the (worst-case) generalization gap, the best epoch is not chosen using a validation dataset, as it should be done in practice.

We report the mean and 1-sigma error bars over 5 runs, run on the same splits but with different seeds.

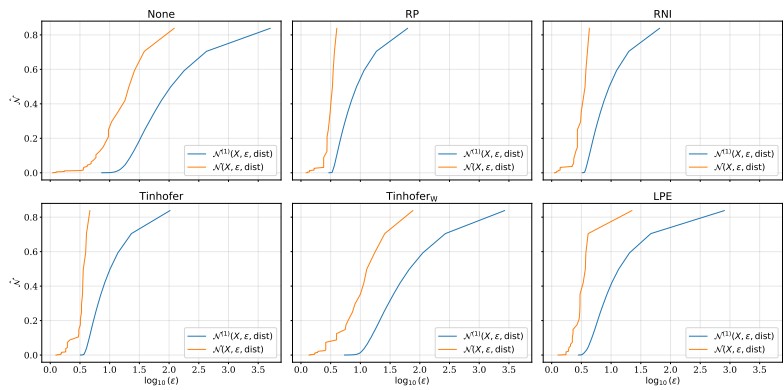

Figure 5: Covering numbers for the NCI dataset

### E.3.4 Covering numbers

For the section on covering numbers, we used four real world datasets for binary classification, as described in Section E.2.

As a metric on the relabeled graphs, we use the Wasserstein distance between the continuous WL embeddings given by [60], which we call WWL distance. In particular, we choose $K_{WL} = 5$ WL iterations to obtain the embeddings, ensuring that $K_{WL}$ is at leas as large as the message passing iterations used in the GNN models. This gives the WWL distance the ability to distinguish all the graphs that the GNNs can distinguish, which is needed to ensure Lipschitzness. In fact, this property cannot be guaranteed, as there are graphs that are not WL-isomorphic but that have WWL distance 0. To mitigate this issue, we assign one-hot encodings with degree centralities to the unlabeled graphs. We then compute $\hat{\mathcal{N}} = \mathcal{N}^{(1)}(\text{Rel}(D), \epsilon, \text{dist})/|D|$ as a proxy for the sample complexity using the WWL distance and $\epsilon = 0.05$. In fact, the covering number we report is an approximation (which is also an upper bound) computed by running a k-median clustering algorithm and selecting the minimum $k$ for which the objective function is less than $\epsilon$.

We use, as the $\text{GNN}_K$ architecture, the one described in Section A.1, with $K = 2, 5$ linear-layer-based message passing layers. We fix dimension of the GNN embeddings to 128. We use the Adam optimizer [32] with learning rate 0.005 and batch size 256.

The model is trained for 1000 epochs. The epoch at which the accuracies are reported is selected as the one with highest training accuracy, with ties broken by the lowest training loss. Note that since the focus of this paper is on the (worst-case) generalization gap, the best epoch is not chosen using a validation dataset, as it should be done in practice. We split the dataset in train and test by assigning 90% of the graphs to the test set at random for each run.

We report the mean and 1-sigma error bars for the accuracies over 5 runs with different seeds. We report the covering number just for the first seed due to the high computational costs involved with the calculation.

## F  Additional results on covering numbers

In this section, we report additional results on the covering numbers on real world datasets. In particular, Figures 5, 6, 7 and 8 report the covering number curves for the NCI1, Mutagenicity, IMDB-b and COLLAB-b datasets, respectively. The results are obtained from the train-test splits of the first run, as described in Section E.3.

We plot both $\hat{\mathcal{N}} = \mathcal{N}^{(1)}(\text{Rel}(D), \epsilon, \text{dist})/|D|$, which depends on the 1-norm covering number, and $\mathcal{N}(\text{Rel}(D), \epsilon, \text{dist})/|D|$, which depends on the ($\infty$-norm) covering number of the relabeled dataset. We observe that the 1-norm covering numbers are significantly lower than the $\infty$-norm covering numbers across all scales and datasets, providing experimental evidence of the importance of the results of Lemma 2.

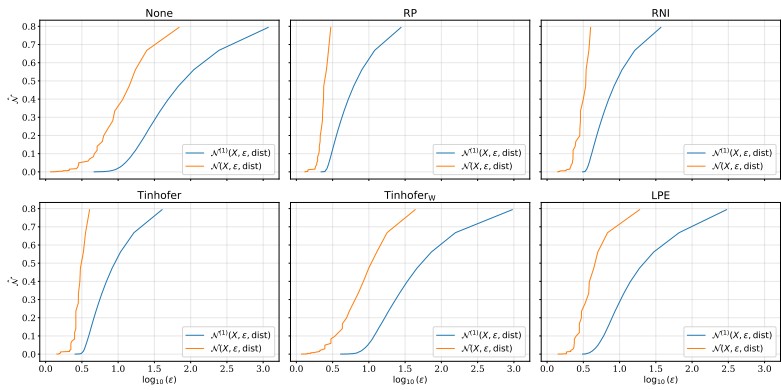

Figure 6: Covering numbers for the Mutagenicity dataset

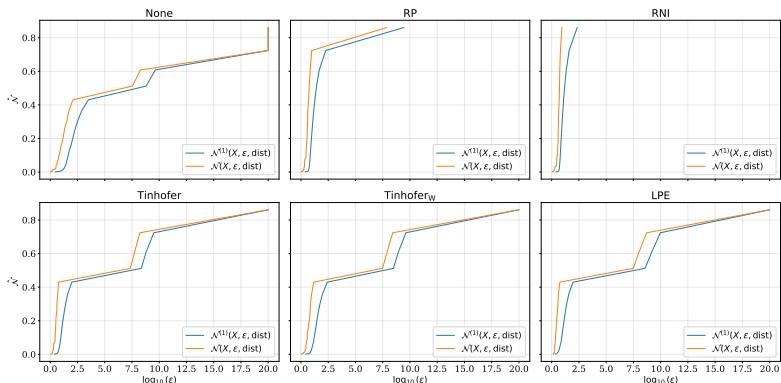

Figure 7: Covering numbers for the IMDB-b dataset

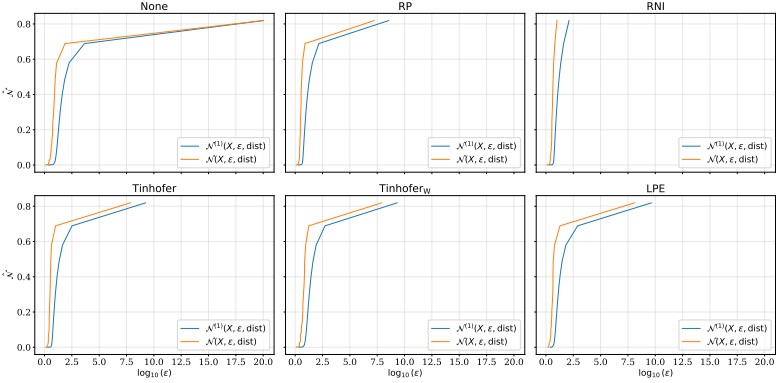

Figure 8: Covering numbers for the COLLAB-b dataset

