# OpenReview forum: "On the Expressivity and Sample Complexity of Node-Individualized Graph Neural Networks"
_NeurIPS.cc/2024/Conference — NeurIPS 2024 poster_

### Official Review · Reviewer_Tve2 · 2024-07-10

**Soundness:** 3
**Presentation:** 3
**Contribution:** 4
**Rating:** 7
**Confidence:** 3

**Summary:**

The authors explore node individualization schemes and argue that they can improve the expressiveness of shallow GNNs and provide bounds on the sample complexity of these methods. This allows node individualization schemes to be compared in this context. The theoretical findings are then substantiated with experiments.

**Strengths:**

- Theoretically comparing different node individualization schemes in terms of their generalization abilities is an extremely important result for the community and can help with future architectural design decisions.
- Theoretical results involving sample complexity are well explained and validated experimentally and the choice of Tinhofer individualization is well motivated and validated.

**Weaknesses:**

- line [158]. The authors argue that node individualization can improve expressivity for shallow GNNs (with k-weakly individualized graphs). This argument would be more compelling if in practice it is shown that their exists graphs that need a large number of layers to be distinguished. In real-world datasets, particularly with node features, it would be interesting to know if this is ever the case. I understand why this results suggests that GNNs can be better distinguished with fewer layers but I am not convinced (yet) that this occurs in practice.
- Line [307]. You seem to be suggesting that we may only want to label necessary nodes as it is 'potentially more stable to input perturbations'. I don't fully understand this and the way you phase it is not very compelling. Can you expand on this or provide some intuition? I think it would benefit the paper having a result that suggests we may only want to label a fraction of the nodes and why this would be more beneficial than labeling all nodes in some cases.

**Questions:**

only what's mentioned in weaknesses.

**Limitations:**

The authors have listed some limitations in the appendix. I don't see the Lipschitzness assumption as a huge limitation as this assumption is made in many works on GNNs.

---

> ### Author Rebuttal · Authors · 2024-08-06
>
> Thank you for recognizing the novelty and the importance of our work.
> We also would like to thank you for pointing out some potential weaknesses of our paper that we didn't spot before. We are confident that both of them can be addressed by adding a discussion in the paper, which we will do by making use of the extra page granted for the camera-ready version. We think that thanks to your comments, we can make our paper both more impactful (by showing datasets where many layers are needed in practice) and more readable (by providing more intuitions). Please find the detailed comments below.
>
> We hope that our response adequately addresses all your comments and positively influences your final score. Please let us know if you have any further or follow-up questions so that we can try to address them.
>
> > **Weaknesses:**
> >
> > Concerning line [158]
>
> We agree with your observation that, on most real-world graph datasets, relatively few GNN layers would be sufficient to distinguish all pairs of graphs. However, motivated by your comment, we looked for and found real-world datasets where few layers are not enough to distinguish all graphs. This makes indeed for a very compelling argument in favor of our paper.
> In the following, we report the minimum number of WL iterations needed to distinguish all non-WL-equivalent graph pairs for the datasets used in our paper as well as the two additional datasets ${\tt MCF-7}$ and ${\tt Peptites-func}$.
>
> | Dataset          | WL iterations needed |
> | --------------- | --------- |
> | COLLAB | 1 |
> | IMDB-BINARY | 1 |
> | MCF-7 | 6 |
> | Mutagenicity | 3 |
> | NCI1 | 3 |
> | Peptites-func | 6 |
>
> We found that there are quite a few datasets which require at least six WL iterations (when incorporating node attributes). Considering that most GNN architectures commonly use 3-4 layers, we would argue that there are indeed cases where our approach is beneficial in real-world scenarios.
>
> Moreover, we notice that, even though GNNs are theoretically equivalent to WL, in practice they often have lower distinguishing power (e.g. due to bad convergence). We particularly notice this behaviour in the experiments in Section 5.1 (although on synthetic graphs). In fact, this behaviour has been observed in the literature (see, e.g., the recent paper [Wang et al, "An Empirical Study of Realized GNN Expressiveness", in ICML 2024]).
>
> Similar to the experiments in Section 5.1, we now provide further experiments comparing the performance of ordinary GNNs with that of GNNs endowed with ${\rm Tinhofer}$ on the above mentioned datasets ${\tt MCF-7}$ and ${\tt Peptites-func}$. To highlight the difference in performance between the two methods, we extracted a small subset of graphs from both datasets, for which a maximum number of layers (i.e., up to 6 layers) is necessary to distinguish all pairs of graphs. The learning task is to assign each graph to their isomorphism class. Figure 2 in the appended PDF file plots the accuracy of the two methods (i.e., ordinary GNNs with 6 layers and GNNs endowed with ${\rm Tinhofer}$) over 1000 epochs on these two datasets. It can be seen that the GNN with ${\rm Tinhofer}$ converges rapidly while the oridnary GNN in many cases does not even reach 100% accuracy within 1000 epochs; thus confirming the above claim on the relevance of our approch to real-world scenarios.
>
> > Concerning line [307]
>
> Thank you for pointing out the inaccuracy in this paragraph. We will rephrase the sentence and add the following motivating example to give more intuition on the results of Theorem 4 and how they can lead to the design of new relabeling schemes.
>
> Consider, as a motivating example, the two graphs
> in Figure 1 (*Panel (a)*) of the Supplementary PDF to the rebuttal.
> Here, the letters indicate graphs' node labels, and correspond in fact with WL color classes.
>
> The ${\rm Tinhofer}$ scheme would find a canonical ordering on the two graphs by assigning an identifier (e.g., Â) to one of the two nodes labeled with A. Then, assuming a total order A < Â < B < C < D < E, it would concatenate the position of the node in the ordering to the node label. We then obtain (up to ordering of the two A-labeled nodes), the graphs in *Panel (b)*.
> We therefore have that the relabeled graphs have edit distance 3, even though the edit distance of the original graphs was just 1. By making the "tail" of the graphs longer, one can make the difference in edit distances arbitrarily large.
>
> The ${\rm Tinhofer_W}$ scheme would use the Tinhofer algorithm to obtain the same node ordering. The main difference lies in the fact that the scheme appends the position of the node in the ordering within its WL color class. One would then obtain the graphs in *Panel \(c\)*.
> The edit distance thus remains 1, as in the original graphs. Then, by Theorem 4, we argue that the ${\rm Tinhofer_W}$ scheme leads to better generalization. This is indeed validated by the experiments in Section 5.4.
>
> We note that ${\rm Tinhofer_W}$ does not necessarily yield a lower sample complexity on all graph distributions. Therefore we do not provide a further theoretical analysis of this relabeling scheme. However, while our article is primarily focused on the sample complexity bounds, we hope that it will spark several works on new relabeling schemes that guarantee lower sample complexity.

---

> > ### Comment · Reviewer_Tve2 · 2024-08-08
> >
> > Thank you very much for your detailed response to my questions. Indeed, I think these additional discussions make the paper stronger and I have increased my score accordingly and will push for acceptance.

---

> > > ### Author Response · Authors · 2024-08-13
> > >
> > > Thank you very much for your answer and for appreciating our work. We really value your input, which will help us enhance and strengthen our contribution.

---

### Official Review · Reviewer_xW3P · 2024-07-12

**Soundness:** 2
**Presentation:** 1
**Contribution:** 2
**Rating:** 4
**Confidence:** 3

**Summary:**

In this paper, the authors investigate the generalization properties of node-individualized Graph Neural Networks (GNNs). Specifically, they aim to differentiate between various individualization schemes based on their generalization properties. To achieve this, they employ two techniques: VC dimension and covering numbers. Furthermore, they propose a new individualization scheme informed by their findings.

**Strengths:**

**S1 Highly Relevant Topic:** The study of the generalization properties of GNNs is highly pertinent to understanding their capabilities. Investigating this within the context of individualization is particularly interesting, as individualization has been demonstrated to enhance expressive power.

**Weaknesses:**

**W1 Presentation:** The presentation of the results and methods is not clear. Particularly in the section following Theorem 1, where bounds on the number of equivalences are presented across various settings without sufficient explanation. More detail or intuition would be beneficial, especially regarding the Tinhofer individualization. Additionally, the comparison of different schemes based on these bounds is not clearly articulated. For instance, why are only amenable graphs considered for Tinhofer, but not for the other two cases? The discussion on positional and structural encodings also lacks clarity. Statements such as the Laplacian increasing VC dimension require more justification. The segments about randomness and super-exponentiality need further elaboration.

**W2 EGOnet Individualization:** In section 4.2, the authors mention that their proposed 1-layer GNN on individualized egonets is guided by observations on VC dimension bounds. This point is unclear and should be detailed more explicitly. It might refer to the last sentence in that section, but the explanation is insufficient. Clear comparisons between different schemes in terms of VC bounds are necessary.

**W3 Covering Number Result:** The conclusions drawn after Theorem 4 are not very clear. The presentation in this part is suboptimal, lacking clarity. The authors should more clearly indicate the insights gained from these results. Also, the proofs are very simple.

**W4 New individualization schema:** The authors mention that a contribution is the design of a new schema, yet it is hidden well in the discussion at the end of Section 4. There one finds a high level and unclear description of that new scheme (appendix contains more details but haven’t checked that). At least, this major contribution could have been explained in the main paper?

**Questions:**

Please address the weak points.

**Limitations:**

no comments.

---

> ### Author Rebuttal · Authors · 2024-08-05
>
> We thank you for your insights on how the paper could be improved, and for acknowledging the high relevance of our work. We think that, by addressing your comments on the lack of clarity of some sections, the paper will be indeed easier to understand. Please find below both the answers to your comments, and the proposed additional explanations and examples to increase the paper's comprehensibility.
>
> We are however surprised by your evaluation with a score of 3, which entails technical flaws, weak evaluation or inadequate reproducibility.
> We hope that our response addresses your concerns with clarity, and we would therefore highly appreciate if you would consider reevaluating your final assessment of our article.
> We’d be happy to address any additional questions you may have.
>
> >**Weaknesses:**
>
> > W1 Presentation
>
> Thank you for pointing out some sections that lack clarity. Note though that the level of detail that we could integrate into the main paper was limited by the strict page limit. If the paper is accepted, we will use the extra page granted for the camera ready to explain the sections you deem unclear in greater detail and include parts of the Appendix.
> In particular, we will describe the individualization schemes in more detail (see also the reply to Point 7 of reviewer 1).
> For the Tinhofer scheme, we first give a general bound to the size of ${\rm Rel}(G)$ as for the other schemes. We then give a tighter result for the class of WL-amenable graphs. This result holds, as stated in the paper, just for the Tinhofer scheme, so we don't give similar results for the other schemes.
>
> Concerning the Laplacian increasing the VC dimension, we thank you for pointing out that the sentence needs more justification.
> We will replace line 223 as follows:
> "The Laplacian positional encoding [13] is known not to be equivariant in general [52], i.e., there are graphs $G$ such that ${\rm Rel}(G, \omega) \not\simeq {\rm Rel}(G, \omega')$ for some $\omega \neq \omega'$. Therefore we have that $|{\rm Rel}(\mathcal{G}) / \simeq_{WL}| > |\mathcal{G} / \simeq_{WL}|$, which, by Theorem 1, leads to an increase in the VC dimension..."
>
>
>
> > W2 EGOnet Individualization
>
> We think that by incorporating your comment we can make this section indeed more clear. Please find below an answer to your comment and how we would address it in the paper.
>
> In general, we showcase that VC dimension bounds can lead to the development of better architectures tailored to specific tasks.
> In particular, as stated in the paper, we design an architecture whose VC dimension depends on $\mathcal G_\Delta$, which is much smaller than $\mathcal{G}$. This is briefly discussed, as you correctly point out, in the last paragraph of 4.2.
> We would add the following discussion to make this point more explicit.
> "In general, $\mathcal G_\Delta$ is much smaller compared to $\mathcal{G}$, especially for small $\Delta$. This leads to the fact that, in general, ${\rm Rel}(\mathcal G_\Delta)$ will be much smaller than ${\rm Rel}(\mathcal{G})$. Thanks to Thm.1, we then have that the VC dimension of $GNN_{\Delta}^{ego, Rel}$ is in general lower compared to the one of $GNN_{K}^{bin} \circ {\rm Rel}$. This theoretical result is also experimentally validated in Section 5.3."
>
> > W3 Covering Number Result
>
> We point out that, although the proofs are not extremely complex, we are the first to develop these bounds, which we believe could be of interest to the entire statistical learning theory community, even beyond graph learning.
> Moreover, we are the first to apply them to relabeling schemes for graph neural networks.
> Finally, we agree that we can give more intuition on the results of Theorem 4 and how they can lead to the design of new relabeling schemes. Please see the reply to your point W4 for an example that we will include in the paper.
>
>
> > W4 New individualization schema
>
> Please note that we don't claim that the new individualization scheme is a major contribution of the paper. The major contributions are rather the sample complexity bounds, we will make this clearer.
> Based on the covering number bounds, we indeed develop the ${\rm Tinhofer}_W$ scheme.
>
> We once again believe that addressing your comments by expanding this section will make the paper easier to understand. We will use the extra space in the camera-ready to integrate the formal description of the ${\rm Tinhofer}_W$ scheme, which we now give only in the Appendix due to space constraints.
> We will moreover integrate the following intuition on why the ${\rm Tinhofer}_W$ scheme can lead to better generalization (as shown empirically in Section 5.4).
>
>
>
> Consider, as a motivating example, the two graphs
> in Figure 1 (*Panel (a)*) of the Supplementary PDF to the rebuttal.
> Here, the letters indicate graphs' node labels, and correspond in fact with WL color classes.
>
> The ${\rm Tinhofer}$ scheme would find a canonical ordering on the two graphs by assigning an identifier (e.g., Â) to one of the two nodes labeled with A. Then, assuming a total order A < Â < B < C < D < E, it would concatenate the position of the node in the ordering to the node label. We then obtain (up to ordering of the two A-labeled nodes), the graphs in *Panel (b)*.
> We therefore have that the relabeled graphs have edit distance 3, even though the edit distance of the original graphs was just 1. By making the "tail" of the graphs longer, one can make the difference in edit distances arbitrarily large.
>
> The ${\rm Tinhofer_W}$ scheme would use the Tinhofer algorithm to obtain the same node ordering. The main difference lies in the fact that the scheme appends the position of the node in the ordering within its WL color class. One would then obtain the graphs in *Panel \(c\)*.
> The edit distance thus remains 1, as in the original graphs. Then, by Theorem 4, we argue that the ${\rm Tinhofer_W}$ scheme can lead to better generalization. This is indeed validated by the experiments in Section 5.4.

---

> > ### Comment · Reviewer_xW3P · 2024-08-09
> > **Thanks for the rebuttal**
> >
> > Dear Authors,
> >
> > Thank you for your detailed rebuttal and thoughtful responses to my concerns. I appreciate the depth of analysis provided and recognize the value of your work. However, my current score reflects the difficulty I had in digesting the presentation of your concepts, even though I am familiar with most of them. I found several parts lacked clarity, a concern also shared by review QJ5E. I believe your work has significant potential, but I am unable to recommend acceptance in its current form.

---

> > > ### Author Response · Authors · 2024-08-13
> > >
> > > Thanks for your answer. We understand and respect your decision. In any case, we will make sure to use your feedback to make the paper easy to understand.

---

### Official Review · Reviewer_QJ5E · 2024-07-14

**Soundness:** 2
**Presentation:** 2
**Contribution:** 3
**Rating:** 6
**Confidence:** 4

**Summary:**

This paper proposes sample complexity bounds for message-passing graph neural networks (GNNs) with node individualization schemes, i.e., the assignment of unique identifiers to nodes. The authors first introduce a mathematical framework which describes node individualization schemes as a relabeling process. Based on this framework, they provide VC-based sample complexity bounds for several node individualization schemes and a more general VC-based sample complexity bound for substructure identification based on ego nets. The authors then use covering numbers to bound the empirical Rademacher complexity and, building upon this bound, propose a novel node individualization scheme. Finally, they perform experiments on synthetic and real-world datasets for substructure identification and graph classification to evaluate their theoretical results.

**Strengths:**

To the best of my knowledge, this paper is the first to introduce sample complexity bounds for graph neural networks with unique node identification. I think this contribution fills a gap in the current literature and is of interest to the graph learning community. Additionally, the authors propose an individualization scheme based on the Tinhofer algorithm and ego network construction with impressive results on substructure identification tasks.

**Weaknesses:**

While the introduction and motivation are well-written, the paper has some issues with respect to clarity. Overall, it would be helpful to point out explicitly which results hold for permutation equivariant (invariant) GNNs, and which are (additionally) universally expressive. Particularly, the role of $\\Omega$ is not fully clear to me: How does it address permutation equivariance, and how does it introduce pseudo-randomness?

In general, the theoretical results could be better situated within related work: How do the node individualization-based bounds differ from GNNs without node individualization? What is the trade-off between expressivity, permutation invariance and generalization? With respect to writing, it would further be helpful to use the definition environment for key terms in the preliminaries to make them easier to locate (e.g., lines 145-147). Additionally, important references such as 1-WL are missing from the preliminaries and line breaks in inequalities or definitions make the paper difficult to follow. Finally, there is ambiguous language, e.g., do node _embeddings_, _features_, and _labels_ refer to the same concept?

Overall, my current score reflects the paper's clarity issues, which made it difficult to fully assess the theoretical results. Please refer to the **Questions** section for more details. If these issues are addressed, I would be willing to raise my score.

Please find some minor remarks and suggestions below:

* Often the comma is missing after "i.e." and "e.g." (e.g., line 20, line 40, line 120)
* Please consider double-checking how you use plural/singular throughout the paper (e.g. GNNs in line 149)
* lines 19-23. Please consider breaking up this super long sentence
* line 22: certain classes of graphs
* line 35: typo in expressivity
* line 38: language sounds off, suggestion: "These features provide information about the graph topology [...]"
* line 40-41: "The amount of training data required for generalization beyong the training data" -> Consider re-writing this sentence
* line 56: universal -> universally
* line 87: tuple of G -> tuple G
* line 109: dot missing
* line 112: "this algorithm" -> "Tinhofer" or "The Tinhofer algorithm"
* Consider removing "hereinafter" from the introduction of abbreviations, e.g., graph-neural networks (GNNs)
* line 137: $\\Omega$ is in $\\mathbb{N}$ but referred to as integer in the text
* line 140: "in memory" sounds a bit odd (if they have different vertex orderings)
* line 246: form -> from
* line 298: number "of" WL
* line 320: explore -> explores
* line 325-326: double "respectively"
* line 360: indiced -> induced
* Consider not having line breaks in formulae, this sometimes hinders readability
* lines 52-61: consider providing links to the relevant sections/theorems in your list of contributions
* line 532: suffice -> suffices
* line 556: arrow points in the wrong direction
* Consider checking how you refer to theorems etc., this is not always consistent (e.g., Prop. 1 is referred to as theorem in the text)
* Please consider numbering equations
* Reference [38] appears to be published in the future (NeurIPS 2024)

**Questions:**

1. Could you elaborate on your definition of an equivariant function (where domain and codomain are both graphs)? How does this differ from an invariant function from graphs to graphs?
2. Line 107: Why is the max operator used to define $k$? If $|V_G| \\neq |V_H$, then the two graphs are not isomorphic as per 1-WL, as there is no bijection between multisets of different sizes.
3. Why is the graph-level readout defined to be in $[0,1]$ (line 120), but $\\{0,1\\}$ in Prop. 1? Should it be $GNN_{k,\\theta}^{bin}$ in Prop. 1? The same holds for Thm. 2.
4. Could you please elaborate on the additional arguments from $\\Omega$, both with respect to ensuring permutation equivariance (cf. lines 136-141) as well as modelling pseudo-randomness? How do you obtain/choose $\\omega \\in \\Omega$? Does it serve as an enumeration of all possible permutations (and thus is factorial in the number of nodes of a graph)?
5. On a related note, could you be more explicit about what node individualization schemes are permutation equivariant (invariant) *and* universally expressive (as stated in the proof for Theorem 2)?
5. lines 150-151: Does this mean that GNNs could be used similary to WL in the Tinhofer algorithm? Or could you elaborate how we can use $k$ message passing layers to obtain a node individualized graph representation?
7. lines 202-208: Could you elaborate on the relabeling process? Are the words "features" and "labels" synonymous in this context? If we update the label for each node $v_i$ as $(v_i, i)$, doesn't this make every node set $V_c$ a singleton?
8. Proof of Lemma 1: Shouldn't this be $VC(\\mathcal{G}, GNN_K^{bin}) = |\\mathcal{G}/ \\simeq_{WL} |$ (instead of $|\\mathcal{G}/ \\simeq |$)?
9. All results hold for bounded graphs only. Can you say anything about unbounded graphs?
10. What is the computational complexity of $GNN^{ego}_{\\Delta_P}$ with the Tinhofer individualization scheme?

**Limitations:**

All presented results hold only for bounded graphs. Additionally, as the authors remark themselves, the VC bounds are somewhat vacuous, as we would have to sample from the set of all possible relabeled graphs.

---

> ### Author Rebuttal · Authors · 2024-08-04
>
> Thank you for recognizing the novelty and the potential impact of our work to the graph learning community.
> Moreover, we want to thank you for your extremely thorough and insightful review, which is certainly not to be taken for granted.
> Your comments made us spot and address some sections of the manuscript that required further clarifications, therefore enhancing the paper's quality and readability.
> We believe that we can address most of your concerns and questions with small edits to the paper, possibly using the extra page granted in the camera ready version.
>
> Please find detailed answers to all of your questions below.
>
> > In general, the theoretical results could...
>
> Our sample complexity bounds are applicable even without node initializations (by choosing the relabeling function to be the identity), leading to the bounds by [31 (Morris et al. 2023)] (for bounded graphs).  We will make this explicit.
>
> We will include the reference to 1-WL and use the extra page to avoid line breaks.
>
> Features and node labels are indeed used as synonyms, and we will change the wording to use just node labels (see also question 7). Embeddings have a similar meaning but are the output of a GNN layer, as opposed to node labels, which are intrinsic properties of the input graph. We will make all of this explicit.
>
> > Minor remarks
>
> Thank you, we will fix all typos.
>
> > **Questions:**
>
> 1. Our definition of equivariance is slightly relaxed with respect to some works in the literature, but it is still sufficient for our purposes (line 138, line 220), and avoids making the notation heavier. We will mention this explicitly in the definition.
> For what concerns invariance, note that we use the term just for real-valued functions.
> For functions outputting graphs, a common definition would be that $G \simeq H$ implies $f(G) = f(H)$ (equality, not isomorphism). Notice that such a function would be still called  equivariant under our definition.
>
> 2. The max operator is indeed not necessary. We will replace the sentence with "...if it holds for $k = |V_G| = |V_H|$".
>
> 3. In line 120 we define the output to be continuous in accordance with the literature [34 (Morris et al. 2019)], and it can be interpreted as the probability of $y$ being 1. In Prop.1 and Thm.2 the function $f$ to be realized takes binary values, and technically it could be represented by a continuous-output-GNN (with no output in $]0,1[$). We agree with you though that this could be unclear. We then propose the following changes:
>  - In Prop.1,  we will use $GNN_{k, \theta}^{\rm bin}$ as you suggest.
>  - In line 249, we will change $GNN_{1,\theta}$ to $GNN_{1,\theta}^{\rm bin}$, so that both $h_v^{\rm ego}$ and $h_G$ belong to $\{ 0,1 \}$.
>
> 4. The choice of encapsulating the pseudo-randomness of models in an additional parameter is needed to model the sample complexity, and is used for example also in [17 (Franks et al. 2021)].
> As you correctly point out, it could serve as an enumeration of all possible permutations of a graph and its size can be therefore factorial in the number of nodes.
> We'd like to highlight that the choice of $\omega \in \Omega$ is never done explicitly, and we just use the size of $\Omega$ (or an upper bound to it) to bound the VC dimension of the hypothesis class, such as in the results of Lemmas 3, 4 and 5 in the Appendix.
>
>
> 5. By Prop. 1, all node indiv. schemes are universally expressive. Note that not all relabeling schemes are.
> As mentioned in line 220 some relabeling schemes, such as RWPE, are deterministic and permutation equivariant, but they cannot guarantee universality in general.
> A node indiv. scheme that is at the same time universally expressive and permutation equivariant can be obtained, e.g., by adding a canonical labelling to the nodes. This choice though is highly impractical due to the complexity of such algorithms. We then use the Tinhofer algorithm, that on WL-amenable graphs is canonical, and therefore guarantees both universal expressivity and permutation equivariance (Lemma 1).
>
> 6. Exactly. The first $k-1$ layers of the GNN would be used to simulate WL on the graph. This is possible due to the results of [34 (Morris et al. 2019)].
>
> 7. Features should be replaced with labels. Moreover, we think that you could have interpreted the sentence after "in fact" as a continuation of the algorithm for relabeling, while it is an alternative version of the algorithm. Re-phrasing:
> In the simplest version of the RP scheme, we update the label for each node $v_i$ as $\ell(v_i) = (\ell(v_i), i)$.
> The second version of the scheme instead first partitions the nodes based on their labels into ${V_1, \dots, V_C}$. Then, letting $V_c = (v_{i_1}, \dots, v_{i_{|V_c|}})$, it updates the label for each node as $\ell(v_{i_j}) = (\ell(v_{i_j}), j)$.
>
>
> 8. For the set of WL-amenable graphs, $|\mathcal{G} /\simeq_{WL}| = |\mathcal{G} /\simeq|$. We will make this explicit.
>
> 9. The assumption of bounded graphs is indeed a limitation. However, this assumption is reasonable for many real-world graphs. For example, drug-like molecules generally have less than hundreds of nodes.
> Secondly, there are very few results for unbounded graphs in the literature. Notably, [31 (Morris et al., 2021)] achieves VC bounds for unbounded graphs, but they have to assume that the number of WL color classes is bounded.
> In this case, one can probably extend the results for GNNs with relabeling schemes (not for node indiv. schemes, which break the regularity of the graphs by design), and this is an interesting question for future work.
>
> 10. Let $n_E, m_E$ be the max number of nodes and edges of a ego-net and $n, m$ the number of nodes and edges in the original graph. Extracting each egonet, e.g., using a BFS, is $O(n \cdot m_E)$. Let $T=o(n_E \cdot m_E \log n_E)$ be the max time to run Tinhofer on an egonet, then it takes $O(n T)$ to run the scheme.
> Running one layer of a GNN on all ego nets will take time $O(n \cdot m_E)$. We will mention this.

---

> > ### Comment · Reviewer_QJ5E · 2024-08-12
> >
> > I thank the authors for their detailed response and the additional experiments and examples.
> >
> > Two quick notes on the rebuttal:
> >
> > * A3: Sounds good to me.
> > * A4: Thank you for the clarification. I think this needs to be made much more explicit in the paper. I am also still struggling with the term ''pseudo-randomness''; perhaps you could explain that $\omega \in \Omega$ can be seen as a random seed (if I understood the role of $\Omega$ correctly now)?
> >
> > I also read the other reviews as well as the authors' rebuttals and decided to **raise my score**. While I agree with reviewer xW3P on clarity issues, given the detailed and thoughtful rebuttal by the authors, I am positive that this could be improved for the camera-ready version.

---

> > > ### Author Response · Authors · 2024-08-13
> > >
> > > Thank you for your positive feedback. Once again, we sincerely appreciate the detailed review and helpful suggestions you provided, which motivated us to put an extra effort to ensure that we could deliver the best version possible of the paper.
> > > Concerning question A4: Indeed $\Omega$ can be understood as a random seed. We will mention it.

---

### Author Rebuttal · Authors · 2024-08-04

Dear reviewers,

we would like to express our appreciation for the positive comments on our paper, recognizing
- the novelty of our work;
- the strong significance of our work for the community, potentially guiding future architectural design decisions;
- the strength of experimental results, particularly on substructure identification tasks.

We would like to thank you for the detailed and insightful feedback, which we believe have strengthened our paper considerably. Below, we provide detailed responses to your comments. Please also refer to the appended PDF file for further experimental evaluations highlighting the relevance of our approach in real-world scenarios as well as a motivating example for the ${\rm Tinhofer_W}$ relabeling scheme.

---

### Decision · Program_Chairs · 2024-09-25

**Decision:**

Accept (poster)

**Comment:**

The paper focuses on an important and well-studied problem, the design and expressive power and generalization of graph neural network (GNN) architectures. The work provides a novel analysis of the sample complexity of GNNs enhanced with node individualization schemes, offering insights into their expressiveness and generalization capabilities. All reviewers agreed that the topic is of high importance and that the analysis presents a valuable contribution to the community. The theoretical findings are backed by numerical experiments.

While there are concerns about the scope of the experimental validation, particularly the reliance on synthetic datasets and certain criticized node classification datasets, the consistency of the results with the theoretical predictions adds credibility to the work. Additionally, although the presentation has been noted as somewhat challenging to follow, the underlying contributions are significant and warrant acceptance. The paper would benefit from further improvements in clarity, but these issues do not detract from the overall impact and novelty of the research.

The authors are expected to take into account the author comments to improve the accessibility and clarity of the work.